# The People versus Behavioral Science: Alignment between lay and scientific understanding of compliance

Christopher P. Reinders Folmer[1]*, Malouke E. Kuiper[2], Benjamin van Rooij[1,3]

1 Amsterdam Law School, Center for Law and Behavior, University of Amsterdam, Amsterdam, the Netherlands, 2 Department of Law, Society & Crime, Erasmus School of Law, Erasmus University Rotterdam, Rotterdam, the Netherlands, 3 School of Law, University of California, Irvine, California, United States of America

* c.p.reindersfolmer@uva.nl

## Abstract

Since the beginning of the 21st century, behavioral insights have increasingly been used to inform policy. However, rather than land in a vacuum, such insights arrive in a setting where the citizens they target themselves have lay understanding of what shapes behavior—an understanding that may critically shape their endorsement of such policies. As such, to change behavior through behavioral science, people's lay understanding is of pivotal importance. Yet so far, we know little about how laypersons think about changing behavior, or whether their lay understanding of this aligns with that of science. The present research examines this question by focusing on rule compliance, a frequent target of behavioral influence. In a large-scale survey (N = 3326), we assess participants' lay perceptions of how compliance is shaped by different mechanisms identified in the academic literature, and compare this to empirically observed associations. For this purpose, we focused on behavioral measures against COVID-19—newly introduced rules that applied universally, where behavioral insights were widely utilized to promote compliance, and ample empirical data exists on relevant mechanisms. Our findings revealed that at the aggregate level, there was substantial concordance between participants' lay understanding of how different mechanisms contributed to compliance and empirical evidence—although participants seem to overestimate their impact in absolute terms, and underestimate their variability. While this result need not imply that laypersons individually have an accurate understanding of what shapes compliance, it does suggest that collectively, people's lay understanding of these processes was broadly aligned with a scientific understanding. Moreover, their lay understanding of these processes was associated with their preferences for policies to shape compliance by leveraging these mechanisms. In this way, people's lay understanding may provide a stepping stone for scientifically informed policies—and an important reservoir of experiential knowledge that the science of behavior change can tap into.

**Data availability statement:** All data and syntax files are publicly available via the University of Amsterdam's Figshare repository, and can be accessed via https://doi.org/10.21942/uva.29154500.

**Funding:** This project was funded by a grant from ZonMw, the Netherlands Organisation for Health Research and Development (grant number 10430022010017), and by the European Research Council (ERC) under the European Union's Horizon 2020 research and innovation programme (grant agreement no. 817680). The funders had no role in study design, data collection and analysis, decision to publish, or preparation of the manuscript.

**Competing interests:** The authors have declared that no competing interests exist.

## Introduction

Since the beginning of the 21st century, a behavioral revolution has taken place, and has fundamentally impacted many areas of governance, public policy, and the business world [1,2]. Social scientific insights about how behavior is shaped by incentives [e.g., 3], social norms [4], cognition [5], and other mechanisms have been popularized by books like Thinking, Fast and Slow [6] and Nudge [7]. These and other insights from behavioral science have increasingly influenced the way that these actors understand behavior in their respective domains, as well as the ways in which they seek to influence it—for example by leveraging social norms, choice architecture, and predictable irrationalities to shape better behavior, instead of traditional instruments like creating more rules, punishment, and liability [8–13]. Governments and companies around the world have created behavioral insight teams which translate such knowledge into practical interventions [8,14–16]. In this way, behavioral insights have increasingly informed policy [17–19].

Important to realize, however, is that scientific insights about behavior, as well as the policies that they inform, do not land in a vacuum. Rather, they arrive in a setting where the citizens they target themselves have lay knowledge and understanding about behavior and what shapes it. This understanding is rooted in their intuitive comprehension of these processes, as well as in the experiential knowledge that they have developed through their personal experiences [20–24]—for example by influencing others and being influenced themselves. This lay understanding, however, may not align with scientific evidence, or with policies based on this [e.g., 25, 26–29]. And if citizens' lay understanding does not align with scientifically informed policies, then this may have important repercussions for their response to these. Indeed, differences between lay and scientific understanding may not just lead to the rejection of scientific evidence, but also to opposition of policies based on scientific insight [30–34], such that science-based policies may have less popular support, and are more difficult to develop and maintain. In sum, for the scope of changing behavior through insights from science, people's lay understanding is of pivotal importance.

So far, however, little is known of citizens' lay understanding of behavioral change, or the extent to which this may align with what behavioral science shows about this. While research has examined laypeople's knowledge about specific controversial topics (like climate change, vaccination, evolution, and GM foods), as well as their acceptance of scientific evidence about this [35–38], little research so far has examined how laypersons intuitively think about how to change behavior, or which mechanisms they see as being influential for this. Moreover, there is little research that has sought to compare their lay understanding of these processes with what science demonstrates about this. This contrasts with other domains where lay understanding or intuitive beliefs have been compared to scientific evidence, such as psychology [39–42], consumer science [43], health [25,44,45], genetics [46], biology [47,48], geology [49–51], climate science [52,53], physics [54–56], mathematics [57], and economics [26,58–65].

Understanding the fit (or lack thereof) between scientific insights and citizens' lay understanding of behavioral change is essential for several reasons. Firstly, gaining

insight into lay understanding of behavioral change is important for behavioral science in order to identify which of its empirical insights may be more or less intuitive, and hence more or less easily translated to stakeholders who seek to utilize such findings. Lay understanding of behavior change is also important for policy makers, because policies that are mismatched with what their targets believe to be effective may be resisted or rejected. In this way, lay understanding may reveal obstacles to behavior change, as well as possible avenues for overcoming these. Lay understanding may also serve as a tool for evaluating the effectiveness of behavioral change interventions, for example to identify whether targets' beliefs about what works may change after exposure to scientifically informed interventions [e.g., see 66, 67]. Last, lay understanding may bring to light topics that require further empirical study, in light of evidence that counterintuitive findings can lack replicability [e.g., see 68].

The present research therefore examines how citizens' lay understanding of behavioral change may align with behavioral scientific empirical insight into this question. It does so by focusing on a domain where behavioral change is of pivotal importance, namely that of promoting compliance with rules and preventing misbehavior. Compliance is a key issue in contemporary regulatory governance that seeks to safeguard society, the economy and the environment from harm. Regulation is only effective when those targeted by its rules come to comply with them. While there is a vast field of interdisciplinary theoretical and empirical work about why people and organizations comply, there is limited study about whether such science aligns with lay understandings of compliance. This is remarkable, as people in their everyday roles as children, parents, citizens, employees, or leaders have firsthand and secondhand experience with following and breaking rules, and ways of influencing this. To understand whether people's lay understanding of shaping compliance aligns with a scientific understanding, we focus on a unique period where novel behavioral measures were introduced that applied to everyone, where a broad range of behavioral mechanisms were at play, and where the question of what shapes compliant behavior has been extensively researched: the COVID-19 pandemic. We compare empirical evidence on the mechanisms that shaped compliance in this setting with citizens' lay understanding of this. In this way, our research illuminates lay ideas about behavioral change in this setting, and the extent to which these align with empirical evidence about this question. We draw from this conclusions about citizens' lay understanding of behavior, and what this implies for policies informed by behavioral science.

## Lay understanding

While an extensive body of research situated in a multitude of disciplines has examined what shapes compliant behavior [for an overview, see 69], the question how lay citizens think about this has received scant attention. However, this question does connect to several peripherally related literatures, which focus on people's lay understanding of other subjects or processes, as well as its concordance with the way that these are understood by science. While such insights do not directly illuminate people's lay understanding of compliance, or its relation to behavioral scientific evidence, they can provide more general indications of people's lay understanding of everyday phenomena, and its alignment with what science shows about this.

**Lay understanding of other scientific fields.**  The question to what extent laypersons' understanding of particular phenomena may align with that of science has received attention in a broad range of fields, including psychology [24,70–75], consumer science [43,76,77], health [25,44,45], genetics [46], biology [47,48], climate science [52,53], geology [49–51], physics [54–56], mathematics [57], and economics [26,58–65]. Different labels have been used in these literatures, including "commonsense" [24], "folk" [54], "lay" [72], "everyday" [76], and "untutored" [72] knowledge or understanding, as well as "predictions" or "forecasts" [65]. Globally, all refer to the way that laypersons understand or believe (often complex and opaque) processes in the real world to operate, compared to the systematic understanding of these processes in science. While this question has received attention in a diverse range of fields, this is typically limited to no more than a small subset of studies, focusing on a small subset of processes (as opposed to a range of processes operating simultaneously, as is the case in most everyday situations).

 

Insights from these literatures have demonstrated both overlap between lay understanding and scientific understanding, but also substantial differences, According to Heider [70], "the ordinary person has a great and profound understanding of himself and other people which… [is] …unformulated or only vaguely conceived" (pp. 2–4). Indeed, Kelley [24] reviews a range of studies in which laypersons were able to predict with considerable accuracy the operation of a range of social and psychological phenomena as indicated by empirical science, including learning and intelligence [71,72,75,78,79]. Kelley [24] suggests that overlap between lay and scientific understanding thus may be expected in that "principles" derived from scientific research may already be part of common knowledge. Additionally, he observes that overlap may be expected because lay or commonsense understanding of particular processes may also shape the way that scientists think about, and study, these [for a similar argument, see 75,76]. Nonetheless, researchers have also claimed that compared with the scientific understanding, people's intuitive understanding of events in the physical world are likely to lack integration, depth, and systematicity [54,80].

Kelley [24] has hypothesized, reflecting about social and psychological processes, that lay (commonsense) understanding is "most likely to be extensive and valid when it refers to events that exist at a middle (meso) level of complexity" (p. 6), as opposed to events occurring at the micro-level (rapid, small-scale, difficult-to-observe behaviors) or at the macro-level (long time spans, large numbers of people). According to Kelley, the meso-level constitutes the level at which "most of subjective daily life is carried on" (p. 6), and encompasses events involving "individual, goal-directed activity, immediate and direct consequences, time-spans of minutes to days, and face-to-face interaction of small numbers of people" (p. 6). Kelley argues that this level is generally the focus of attention in everyday life, and provides the type of information that is suitable for conscious and deliberate processing. This enables laypersons to develop more elaborate, and potentially more valid commonsense understandings of how such events and processes operate. Furthermore, Kelley hypothesizes such understanding to be more extensive and accurate for events that are familiar (rather than unfamiliar) to people, and which they observe (rather than participate in). In line with this thesis, laypersons have been found to be able to predict with considerable accuracy how effort is shaped by different (familiar and observable) economic and social factors, particularly when the predictions of different individuals are aggregated [65]. Conversely, research has indicated only limited alignment between lay and scientific understanding for granular, unobservable processes for example relating to physics [54,56,80] and memory [81], as well as for large-scale processes relating to climate change [52,53] and the economy [64,82,83]. Kelley's predictions also align with findings on science skepticism, where a significant portion of the population is shown to hold beliefs that oppose the scientific consensus on (micro and macro) processes like climate change, vaccination, and genetic modification [35,84–87].

Nonetheless, this need not imply that lay understanding is otherwise well aligned with scientific understanding, however. For example, Kelley [24] also observes that commonsense understanding of everyday social and psychological processes is rife with contradictions that are logically irreconcilable. Moreover, research has indicated that even for everyday phenomena that people have direct experience with, their lay understanding can differ substantially from a scientific understanding, for example with learning [81,88] and parenting [42]. Conversely, some research has revealed considerable alignment between lay and scientific understanding even for difficult-to-observe processes such as those relating to genetics [46]. In sum, previous research provides mixed evidence on the accuracy of lay understanding compared to a scientific understanding of these processes.

**Lay understanding of compliance.** While we largely lack a specific literature on lay understanding of compliance science, there have been studies that have looked at closely relevant questions. These are mostly situated in the broader literature on lay or "commonsense" psychology, which according to Kelley [24] is focused on understanding "common people's ideas about their own and other persons' behavior and about the antecedents and consequences of that behavior" (p. 4). Research has shown that when observing the social world, people spontaneously and automatically make causal inferences, whereby they infer which factor(s), usually among multiple possible candidates, caused a particular outcome or behavior [89–91]. Globally, such intuitive understandings are formed from an inventory of heuristic

principles or "clues" about causal relationships which are construed based on prior experience and understanding [92]. Through this lens, people come to perceive a particular action as the result of a specific cause, combination, or string of intermediate causes culminating in the observed behavior [90]. Jara-Ettinger et al. [74] more specifically contend that people interpret the observed (intentional) behaviors of others through the lens of a "naïve utility calculus," in which others are assumed to pursue those goals and actions that maximize their expected rewards, relative to the expected costs. By interpreting the actions (or inaction) of others in terms of utility maximization, people are able to infer how people's actions may be shaped by their beliefs and desires, their knowledge and preferences, and their character in light of the features of their (social and physical) environment [74].

While such research provides important insight into the way that laypersons make inferences about particular outcomes and behaviors, it has principally focused on relatively simple situations, as opposed to complex situations in everyday life such as those relating to compliance, in which a multitude of processes may simultaneously be at play. Moreover, such research has focused less on the question to what extent people's lay understanding of these processes may be aligned with empirical evidence reflecting the scientific understanding of these processes. Indeed, although Jara-Ettinger et al. [74] contend that "more often than not we accurately evaluate the behavior of others" (p. 600–601), there is, to our knowledge, no research that empirically substantiates this. Indeed, the few studies that have sought to directly compare people's intuitive ideas about others' behavior with empirical findings have shown mixed results. Epstein and Teraspulsky [93] found that people were able to predict with considerable accuracy how predictive behaviors in one situation were for behavior in different situations relating to the same trait, compared to actually observed correlations for these relationships. Conversely, Stavrova et al. [94] found that people overestimated the impact of personality traits on prosocial behavior, in comparison with their actual observed association with this outcome.

More directly relevant to compliance are literatures that look into lay understanding of processes relating to social influence. A first relevant strand is the literature on everyday persuasion knowledge [76,77], which refers to people's everyday knowledge of persuading others, and of being the target of others' persuasion attempts. A second relevant strand of research concerns research on parenting [39–42]. This is relevant to lay understanding of compliance because raising children inherently involves behavioral change, with parents seeking to teach or promote desired, and prevent or extinguish undesirable behaviors. As such, this is one of the principal domains in which people may have developed an understanding of how compliant behavior is shaped, based on their direct personal experiences (as parents and/or children) or observations of others.

Research on everyday persuasion knowledge has mostly examined this from the perspective of marketing and advertising, and the way that this influences people's preferences and decisions [for a review, see 43]. Particularly relevant is research that has sought to directly measure laypersons' understanding of key principles from this literature [95–97]. Such research indicates that laypersons generally have modest to moderate understanding of persuasion tactics [95,97–100]. However, such findings refer more to their insight into marketing tactics (e.g., effects of price discounts) than to more general insight into effective ways of influencing others and their behavior.

Research on lay understanding of parenting has mostly focused on knowledge of child development, but more recently, research has begun to examine knowledge of effective parenting strategies [39–42]. Such research examines to what extent parents have knowledge of effective parenting practices as identified in scientifically-based and empirically proven parenting programs. In a series of (modest-sized) studies of Australian parents, Winters and colleagues found that these had relatively high levels of knowledge of effective parenting strategies, identifying the correct answer to questions about the promotion of child development, principles of effective parenting, use of assertive discipline, and causes of behavior problems in 78–83% of instances [39–41]. However, such knowledge declined at lower income and education levels, which were represented less most of these studies [39,40]. Conversely, Kirkman et al. [42] conducted a series of larger-scale surveys in (mostly) Australian and American samples, zooming in specifically on knowledge of (evidence-based) parenting strategies for reducing child conduct problems, including encouraging positive behavior, using

discipline, managing high-risk situations and sibling conflict, partner support strategies, and parental self-care. Here, lower levels of knowledge were reported: indeed, participants in the main study indicated the correct answer in only 51.9% of cases—although this could be significantly increased (to 79.6%) after completion of a Behavioral Parent Training program.

There have been a few studies that have looked more directly at lay understanding of science about compliance. Such studies have focused on particular aspects of underlying factors that shape compliance (compliance mechanisms) or particular interventions to enhance compliance (such as punishment, rewards, or honesty oaths). Studies have for instance sought to understand how the public thinks about the root causes of crime. Na and Loftus [101] found that among both American and Korean participants, crime was attributed more to external and societal factors (e.g., upbringing, "flaws in society") than to internal factors (e.g., deliberate choice). However, this study did not zoom in on more specific compliance mechanisms, nor compare these attributions with empirically observed associations. Other studies have looked at whether policy interventions affect compliance. For example, Furnham and Hughes [81] found that over 50% of participants believed that punishment is a highly effective means of changing behavior. Similarly, Vaughan [102] found that 77% of participants believed consistent (rather than intermittent) rewards to be the best way to achieve persistent behavior change after training. In both cases, their lay understanding of these processes was opposed to the existing scientific evidence about these questions. Finally, Zickfeld et al. [103] found that both laypersons and experts failed to predict which honesty oaths were most effective for reducing simulated tax evasion (although experts were less inaccurate). While these insights are few, they do imply that people do hold particular beliefs about what shapes (non)compliant behavior, and that their lay understanding of this may not, or not consistently, align with a scientific understanding of these processes.

**Lay understanding and (scientifically informed) policy.** The alignment between people's lay understanding of what shapes compliance and the scientific understanding of these processes is particularly important because differences between the two may not just lead to the rejection of scientific evidence, but also to opposition of policies based on scientific insight. As a case in point, research has indicated that laypersons' poor understanding and misperceptions of the economy may lead them to favor policies that are likely to have adverse effects, like restrictions to trade or social welfare [29,61]. Similar results have been observed in other domains, such as climate change [85]. In this way, lay understanding may represent an important barrier against scientifically informed policies.

Several processes may explain such findings. Insights on moral coherence suggest that people may question the validity or effectiveness of findings and policies that run against their moral principles [104,105]. It may also be the case that people value different goals than those on which research or policies are oriented, for example by seeing retribution, rather than prevention, as the principal reason for punishing crime [106,107]. Importantly, there are indications that exposing people to scientific insights on such partisan issues may even increase, rather than reduce, this divide. For example, Lord et al. [108] found that partisans increased their support for (or opposition to) capital punishment after being exposed to scientific evidence on the relationship between capital punishment and crime [for similar results for climate change policies, gun control, and nanotechnology, see [38,109,110]. Such findings underline that people's lay understanding may lead them to endorse policies that may be ineffective or even counterproductive according to science. This highlights why people's lay understanding of the processes that shape compliance is crucial for understanding the viability and effectiveness of scientifically informed policies to promote this—as well as the ways in which thepersuasiveness and impact of such policies could be promoted.

**Synthesis.** Taken together, insights on lay understanding in other domains provide some indications that people's lay understanding may globally concord with scientific understanding of these processes. On a substantive level, however, there are also many indications that people's lay understanding may differ from the scientific understanding, or even contradict this. It should be noted, however, that many of these studies concern specialist subjects (such as physics or economics) that laypersons may have little direct experience with, and for which they lack the necessary expertise to understand their operation [61]. Conversely, given that all individuals routinely engage in behavioral influence, and are the target of behavioral influence by others, it may be expected that they are likely to have developed sophisticated

lay understandings of the mechanisms that shape particular types of behavior, including compliance. What those lay understandings may be, however, and to which extent these align with behavioral scientific empirical evidence about these mechanisms, is as of yet unknown.

**Scientific understanding of compliant behavior**

The scientific understanding of compliance is oriented on the question why people obey or break rules. This question has been studied across different academic domains, focusing on different types of rules, different mechanisms, and different interventions that may shape this outcome [69]. As a consequence, the academic literature on this question exists as a patchwork of different theories, existing in distinct, compartmentalized silos, each with their own literatures, methods and findings. Recent work has begun to bring these different theoretical strands together to understand how compliance is shaped by the interplay of these processes [111–115].

Within the academic literature on compliance, five core theoretical approaches can be distinguished, each of which has developed in relative isolation from each other, in different disciplines that focus on different settings and behaviors. They can broadly be listed as rational choice theories [116,117], social theories [118,119], legitimacy theories [120–123], capacity theories [124–126], and opportunity theories [127–129]. Each of these literatures has approached the question why people comply by focusing on its own concepts and variables, within the contexts and settings that are most aligned with its theoretical focus. However, recent research demonstrates that the distinct processes on which these theories focus in fact may operate concomitantly in decisions to comply in many social situations [111–115].

**Rational choice approaches.** Rational choice approaches originate from economics [130] and criminology [117,131]. They hold that (non)compliance is the result of a rational evaluation of costs and benefits, such that people will choose to break the rules if the benefits of not complying (minus its costs) exceed the benefits (minus its cost) of complying.

From a rational choice perspective, a first aspect contributing to compliance therefore is the cost of compliance [132,133], with compliance expected to decrease when the costs of complying increase. The second aspect is the benefits of complying [111,134], with compliance expected to increase as the benefits of compliance become greater.

In line with its calculating view of compliance, a particular focus of rational choice approaches is the role of punishment, and its deterrent effect om offending. General deterrence theory holds that people will comply more when punishment for noncompliance is more certain and severe [130,135,136]—although empirical support for the former is far more convincing than for the latter [137]. Nevertheless, from this theory, compliance may be expected to increase as people perceive the punishment for noncompliance to be more severe and certain.

**Social theories.** Social theories of compliance see compliance as originating in the social context in which people are embedded. They include social norms theories in psychology [118,119,138] and social learning theories in criminology [139,140]. According to such theories, compliance originates in the preferences and behaviors of others in one's environment, to which people attune their own behavior [119,141–143]. On the basis of this perspective, it may therefore be expected that the more that others are seen to comply with rules (or are thought to approve of doing so), the more that people will do so themselves.

**Legitimacy theories.** According to legitimacy theories, compliance is rooted in the perceived legitimacy of rules, authorities, or the law in general. They hold that people will comply more the more that they view the authorities and the rules that they create or enforce as legitimate and fair [120–123,144–146]. This encompasses both substantive legitimacy (i.e., the substance of the rules is in line with one's own morals and preferences) and procedural legitimacy (i.e., the procedures through which the rules were adopted and implemented are seen as fair and just), which both have been shown to predict greater compliance [121,122,146]. However, people may also feel a general sense of duty to obey the law, which can sustain their compliance even when they do not agree with the substance of the rules, and in the absence of external reasons for complying [such as enforcement; 147,148]. In sum, based on legitimacy theories, compliance may be expected to increase the more that people morally agree with the substance of the rules, view them as effective and proportional, or feel a general duty to obey the law.

**Capacity theories.** Capacity theories hold that compliance is shaped by people's capacity to comply with the law. These encompass a number of related ideas which claim that people's personal or practical circumstances may make it easier or more difficult for them to do what rules proscribe, for example because they possess or lack the knowledge, resources, or the discretion that is necessary to do so [149–151]. A similar view is reflected in research that sees compliance as resulting from people's ability to exert self-control to restrain themselves from engaging in prohibited behaviors [125,152–154], or from negative emotional states due to straining circumstances, which people may alleviate through rule breaking [155,156]. According to capacity theories, compliance therefore may be expected to increase the greater that people's capacity for complying is.

**Opportunity theories.** Last, opportunity theories see compliance as shaped by the situation at hand, and the opportunities it provides for offending [127,157–159]. Reflecting this rationale, routine activities theory holds that criminal behavior develops more easily when there are attractive targets which are left undefended to motivated offenders [160–162]. More generally, situational crime prevention theory holds that situations may lower the threshold for illegal behavior, for instance by providing concealment or easy access to tools or techniques needed to break the law [129,163]. From this perspective, compliance therefore should be greater the less that there are opportunities for breaking the law.

## The present research

The present research examines how people's lay understanding of what shapes compliance may align with behavioral scientific insights about this question. More specifically, we examine how the different perspectives about what shapes compliance that have been put forward in behavioral scientific research may be reflected in the understanding that lay citizens have about this. We moreover seek to understand how people's lay understanding of these processes may impact which policy interventions they support to promote compliance, and to which extent these align with what the science indicates. For this purpose, we assess how laypersons evaluate the major mechanisms identified in these approaches in terms of their effectiveness for shaping compliant behavior, and compare this with empirical evidence about their impact. We analyze whether mechanisms that laypersons perceive to be more (or less) influential for shaping compliance may indeed show stronger (or weaker) statistical associations with compliance in empirical data that explores these relationships. Moreover, we examine how their lay understanding of these processes may inform their preferences for policies to shape compliance that leverage these mechanisms—and how this may align with, or differ from, what is indicated by the science.

The present paper applies this approach to lay understanding of compliance through a study of behavioral responses to virus mitigation measures during the COVID-19 pandemic. Governments around the world introduced these measures to curb the spread of the SARS-CoV-2 virus, affecting both individuals and corporations by placing far-reaching restrictions on individual behavior [see 164–166]. This setting is suitable for studying people's lay understanding of what shapes compliance because it concerns the introduction of a novel set of rules which applied to all individuals—as opposed to many other rules, laws or policies which only apply to subsets of individuals or companies, and for which responses have already become habitual. Furthermore, it represents a setting in which core variables from all five major compliance theories have been demonstrated to be at play, and where there is extensive empirical evidence about their associations with compliant behavior [for reviews, see 167,168]. These features make this a setting that is ideally suited for examining lay understanding of compliant behavior, and how this aligns with empirical evidence on this question.

Our research examines this question by means of a survey administered to a large-scale representative sample in the Netherlands, a country where initiatives to change behavior were extensively utilized during the course of the COVID-19 pandemic [169–171]. We assessed participants' lay understanding of what shapes compliance with social distancing measures as part of a longitudinal survey that tracked how the focal mechanisms affected their own social distancing behavior. This approach enables us to directly compare their lay understanding of what shapes compliant behavior with empirical evidence about this question, derived from the same participants (who answer both sets of questions). This

allows for a direct comparison between their lay understanding and an empirical understanding of this question. To do so, we compare participants' (lay) ratings of the influence of the different compliance mechanisms on compliance in this setting with statistical indicators of this association obtained from the same sample, as well as with published associations from the body of empirical studies that has studied compliance with social distancing measures using the same materials [111,113,134,172,173]. This approach is similar to Jennings et al. [174] and DellaVigna et al. [65], who used this to compare participants' intuitive forecasts with quantitative empirical results.

## Method

### Ethical approval and consent

Ethical approval for this project was obtained from the Ethics Committee Law of the University of Amsterdam, on April 3, 2020. All participants provided written consent before participating in the study.

### Participants

Participants were 3,761 Dutch citizens (18 years or older) who were recruited by the Dutch online research panel Motivaction, via the website StemPunt.nu. They participated in the survey between May 6 and 19, 2021, as part of a longitudinal study on compliance with COVID-19 mitigation measures, of which the present survey was the fourth wave. Given that the present manuscript focuses on lay understanding, the longitudinal data on compliance will be discussed in a separate paper. Participants were redirected to Qualtrics to fill out the survey. They were rewarded for their participation with 150 StemPunten (an endowment that can be exchanged for gift vouchers at major Dutch webstores).

Of the sample, participants who did not provide consent, failed to complete the survey, completed it implausibly fast or slow (less than 300 seconds or more than 24 hours, compared to a median time of 739 seconds), or failed to pass two attention checks were excluded. For all other cases, missing data were imputed through multiple imputation (see below). The final imputed sample therefore consisted of 3326 cases.

### Demographic and control variables

The following demographic or control variables were recorded: age, gender, employment status, education level, political orientation, and trust in science [175]. Only a very small subset of our sample considered themselves to be part of an ethnic minority (i.e., less than 5%); for this reason, this variable was not included in the main analysis. Additionally, we asked whether participants provided professional care to COVID-19 patients, and whether they or someone else they knew had a health condition that placed them at increased risk from COVID-19. These variables were either registered by the research panel before the start of the study or were recorded during the first wave of the longitudinal study (October 2020).

### Measuring lay understanding

Thirteen items were used to assess lay understanding of what shapes compliance (see Table 1). They were designed to capture the main mechanisms from each of the five major theoretical approaches to compliance: rational choice theories, social theories, legitimacy theories, capacity theories, and opportunity theories. Their perceptions of the influence of these mechanisms on compliance was measured on 7-point scales (1 = "disagree completely," 7 = "agree completely").

### Measuring policy preferences

To measure participants' preferences for policies to promote compliance, 13 items were solicited (Table 2). These presented a range of policy instruments that directly leveraged the compliance mechanisms that participants had previously rated in terms of their influence on compliance (i.e., the lay understanding measures, see Table 1). Their preferences for these policies was measured on 7-point scales (1 = "disagree completely," 7 = "agree completely").

**Table 1. Perceived influence on compliance of main compliance mechanisms (lay understanding measures).**

| Item<br>Compliance with social distancing rules is influenced by the extent to which citizens: | Compliance mechanism | Theoretical approach |
|---|---|---|
| 1.  Think that this rule has negative effects for their income, work, or social life | Costs of compliance | Rational choice theories |
| 2.  Perceive the coronavirus as a threat to their health and that of others | Benefits of compliance | Rational choice theories |
| 3.  Fear that they will be punished should they not comply with this measure | Deterrence | Rational choice theories |
| 4.  See that most others around them comply with this measure | Descriptive norms | Social theories |
| 5.  Morally believe that people should adhere to this measure | Moral alignment (substantive legitimacy) | Legitimacy theories |
| 6.  Regard this measure as effective and proportional | Perceived effectiveness and proportionality (substantive legitimacy) | Legitimacy theories |
| 7.  Generally feel obliged to obey the law | Duty to obey the law | Legitimacy theories |
| 8.  Are aware of the existence of this measure | Knowing the rules | Capacity theories |
| 9.  Are clear about what this measure demands of them | Understanding the rules | Capacity theories |
| 10.  Are practically capable of keeping a safe distance from others | Practical capacity to comply | Capacity theories |
| 11.  Are capable of controlling their own impulses | Self-control | Capacity theories |
| 12.  Experience negative emotions as a result of this measure (e.g., anger, fear, or loneliness) | Strain | Capacity theories |
| 13.  Have opportunities to still come at an unsafe distance of others | Opportunities for offending | Opportunity theories |

## Compliance and its theoretical mechanisms

In order to compare participants' lay understanding of what shapes compliance in this setting with empirical evidence on this question, our survey also directly assessed participants' own reported compliance with social distancing measures, as well as all main compliance mechanisms as these applied to participants' own present situation. This approach enables us to contrast participants' lay understanding of what shapes compliance with empirical evidence about this question derived from exactly the same setting and sample. These measures were based on prior research on compliance in this setting [111,134,172]. The full list of items is presented in S1 File.

**Compliance measure.**  Nine questions were asked to assess participants' compliance with social distancing measures in different situations (1 = "never," 7 = "always"), based on Reinders Folmer et al. [134]. A "does not apply" option was included in all questions for participants for whom these situations did not apply. Responses were combined into a single scale measure (α = .92), with higher scores indicating greater compliance.

Our survey exploratively measured three additional compliance items on other COVID-19 mitigation measures: whether participants wore a facemask where this was recommended, whether they engaged in self-isolation in case of possible symptoms, and whether they got tested in such instances (1 = yes, 2 = no). Because both our measures of lay understanding and those for the different compliance mechanisms focused on social distancing, we do not discuss these other mitigation measures further in this manuscript. However, in light of their probable associations with social distancing, they were utilized as auxiliary variables for imputing missing values (more details below).

**Compliance mechanisms.**  Our measures of the different compliance mechanisms were based on our previous research [e.g., 111,134,172]. Because these prior manuscripts report them in detail, we discuss them only briefly here. All mechanisms were measured in the current survey wave, except for impulsivity (measured during the first survey wave (October 21–29, 2020), duty to obey the law, and negative emotions (both measured during the third (i.e., previous) survey wave; February 19-March 2, 2021).

For **Rational choice mechanisms**, three mechanisms were measured. Costs of compliance were assessed by means of five items [172]. These asked participants to indicate the likelihood of various negative consequences occurring due to the measures to contain the coronavirus (1 = "extremely unlikely," 7 = "extremely likely;" α = .78).

**Table 2. Preference for policies leveraging main compliance mechanisms to promote compliance (policy preference measures).**

| Item<br>Which of the following instruments would you utilize to ensure that people comply more with social distancing rules: | Compliance mechanism | Theoretical approach |
|---|---|---|
| 1. Ensure that this rule has less negative effects for their income, work, or social life | Costs of compliance | Rational choice theories |
| 2. Emphasize more the threat of the coronavirus to their health or that of others | Benefits of compliance | Rational choice theories |
| 3. Punish people more, to instill more fear of violating this measure | Deterrence | Rational choice theories |
| 4. Show people that most others comply with this measure | Descriptive social norms | Social theories |
| 5. Persuade people more that they morally ought to comply with this rule | Moral alignment (substantive legitimacy) | Legitimacy theories |
| 6. Convince people more that this rule is effective and proportional | Perceived effectiveness and proportionality (substantive legitimacy) | Legitimacy theories |
| 7. Impress more upon people that in general they should obey rules and laws | Duty to obey the law | Legitimacy theories |
| 8. Make people more aware of the existence of this rule | Knowing the rules | Capacity theories |
| 9. Explain better to people what this rule demands | Understanding the rules | Capacity theories |
| 10. Make it practically easier for people to keep a safe distance from others | Practical capacity to comply | Capacity theories |
| 11. Train people to better control their impulses | Self-control | Capacity theories |
| 12. Ensure that people experience less negative emotions as a result of this measure (e.g., anger, fear, or loneliness) | Strain | Capacity theories |
| 13. Make it more difficult for people to still come close to others | Opportunities for offending | Opportunity theories |

Benefits of compliance were, in line with previous research [134], conceptualized in terms of the threat to self and others, given that compliance with mitigation measures is more beneficial the greater the health threat is perceived to be. This was measured with three items [172], which assessed to which extent participants perceived the coronavirus as a major threat to self and others (1 = "strongly disagree," 7 = "strongly agree;" α = .92).

Regarding deterrence, two items were used to measure perceptions of punishment certainty [172] for not keeping a safe distance from others (1 = "extremely improbable," 7 = "extremely probable;" r = .68). One item [172] was used to measure perceptions of punishment severity (1 = "not at all negative," 7 = "very strongly negative").

For **Social mechanisms**, social norms for compliance were measured with nine items (based on Reinders Folmer et al. Social distancing in America [134]). These asked participants to what extent most others around them kept a safe distance from others in each of the nine situations mentioned in our compliance measure (1 = "strongly disagree," 7 = "strongly agree;" α = .93).

For **Legitimacy mechanisms**, three mechanisms were measured. One item [172] assessed moral alignment with social distancing measures (1 = "strongly disagree," 7 = "strongly agree"). Three items were solicited to measure participants' perceived effectiveness and proportionality of these measures (1 = "strongly disagree," 7 = "strongly agree;" α = .89).

Duty to obey the law was measured by means of the 12-item Rule Orientation scale [147,148] (α = .91).

For **Capacity mechanisms,** five mechanisms were measured. One item [134] assessed participants' knowledge of whether social distancing rules currently applied in their area (1 = yes, 2 = no, 3 = don't know; recoded 0 = no or don't know, 1 = yes). Three items [based on 172] assessed understanding of social distancing rules (1 = "extremely unclear;" 7 = "extremely clear;" α = .95).

Nine items [based on 134] were solicited to assess participants' practical capacity to comply with social distancing measures. These asked participants whether they were at this moment capable of keeping a safe distance from others in each of the nine situations mentioned in our compliance measure (1 = "completely disagree," 7 = "completely agree;" α = .85).

To assess participants' self-control, five items taken from the 8-item impulse control subscale from the Weinberger Adjustment Inventory [176] (see [134]; α = .75) were solicited, with higher scores indicating greater impulsivity (and hence lower self-control).

Negative emotions were measured by means of six items [172], which asked to what extent participants felt various negative emotions as a result of the coronavirus (1 = "completely disagree," 7 = "completely agree;" α = .89).

Last, for **Opportunity mechanisms**, opportunity to violate was measured with nine items [based on 134]. These asked participants how often they saw opportunities for coming within an unsafe distance of others, in each of the nine situations mentioned in our compliance measure (1 = "never," 7 = "always;" α = .93).

**Imputation of missing values.** All analyses were conducted in Stata 16.0 SE [177]. Descriptive analyses indicated that nearly all cases featured missing data. The variables with the greatest proportion of missing data were compliance at work (42.1% missing) and in public transport (60.4% missing)—missing data on these variables mostly concerned "does not apply"-responses. For all other variables, less than 7.4% of values were missing. Because our previous research demonstrated strong internal consistency between the compliance items [134], we expected that these missing responses could be estimated from the other compliance items, as well as from additional relevant variables in the dataset, such as, employment (i.e., missing at random, MAR). As such, we imputed missing values by means of multiple imputation [see 178,179]. We relied on multiple imputation by chained equations [MICE, see 180,181]. To accommodate the non-normal distribution of most variables, we used Predictive Mean Matching [PMM, see 182] for continuous variables; binary variables were estimated using logistic regression. In line with recent recommendations, imputation was conducted on the item (rather than scale) level [183]; scale means were computed afterwards based on the imputed data. Full details on the imputation procedure, as well as imputation checks and results for two alternative treatments of the sample are presented in S3 File. Table 3 provides descriptive statistics (imputed data) on the dependent and independent variables that are used in the empirical analysis of how the different compliance mechanisms shape compliance.

**Table 3. Empirical understanding of compliance: Means and standard deviations of dependent and independent variables for empirical analysis of mechanisms shaping compliance.**

| Mechanism | Imputed data (N = 3326)[a] |
|---|---|
| Compliance | 5.58 (1.16) |
| Costs of compliance | 3.13 (1.28) |
| Benefits of compliance (perceived threat) | 5.24 (1.43) |
| Punishment likelihood | 2.83 (1.50) |
| Punishment severity | 4.05 (1.81) |
| Social norms | 4.53 (1.23) |
| Moral alignment | 5.90 (1.36) |
| Perceived effectiveness and proportionality | 5.50 (1.38) |
| Duty to obey the law | 4.66 (1.21) |
| Knowledge | 81.07% |
| Understanding | 6.36 (0.84) |
| Capacity to comply | 5.36 (0.97) |
| Impulsivity | 1.89 (0.76) |
| Negative emotions | 2.78 (1.40) |
| Opportunity to violate | 3.51 (1.43) |
| Age | 54.26 (14.85) |
| Trust in science | 4.01 (0.97) |
| Employed | 53.07% |
| Health risk self | 34.61% |

[a]Standard deviations between parentheses.

# Results

## Lay understanding of compliance

Table 4 provides descriptive statistics (imputed data) on how the different compliance mechanisms shape compliance according to the participants (lay understanding analysis). As shown here, participants saw mechanisms from all theoretical families as influential for shaping compliance, although a fixed-effect (within) regression indicated the level of perceived influence to differ significantly between mechanisms, $F(12, 39710) = 301.30$, $p < .001$. Participants intuitively regarded the benefits of compliance, which in this setting consisted of the harm that mitigation measures seek to prevent, to be the most influential for shaping compliance, while deterrence was seen as least influential.

## Empirical understanding of compliance

Next, we examine the empirically observed associations between compliance and the different compliance mechanisms. For this purpose, we conducted statistical analyses on the responses that participants provided about their own compliance with social distancing measures, as well as those to the measures of the different compliance mechanisms as these applied to their own situation (see Compliance and its Theoretical Mechanisms, in the Method section). We assess the empirical associations between compliance and the different compliance mechanisms by computing bivariate correlations. A supplementary analysis using regression analysis is provided under Supporting information (see S5 File).

   **Correlation analysis.** Bivariate correlations between these variables are displayed in Table 5. Because most correlations were significant due to the large sample size, we rely on the standards formulated by Gignac and Szodorai [184], who classify correlations of about .10, .20, and .30 as respectively small, moderate, and large. By this criterion, eleven behavioral mechanisms showed meaningful associations with compliance: (1) benefits of compliance (rational choice theories), (2) social norms (social theories), (3) moral alignment (legitimacy theories), (4) perceived effectiveness and proportionality (legitimacy theories), (5) duty to obey the law (legitimacy theories), (6) knowledge of the rules (capacity theories), (7) understanding of the rules (capacity theories), (8) practical capacity to comply (capacity theories), (9) impulsivity (self-control from capacity theories), (10) negative emotions (strain from capacity theories), and (11)

**Table 4. Lay understanding of compliance: Perceived associations of compliance mechanisms with compliance according to lay participants.**

| Mechanism | Mean (SD) |
| --- | --- |
| 1. Costs of compliance | 4.80 (1.41) |
| 2. Benefits of compliance (perceived threat) | 5.63[a] (1.26) |
| 3. Punishment | 4.41[b] (1.56) |
| 4. Social norms | 5.38[c] (1.23) |
| 5. Moral alignment | 5.42[d] (1.24) |
| 6. Perceived effectiveness and proportionality | 5.36[e] (1.32) |
| 7. Duty to obey the law | 5.20[f] (1.29) |
| 8. Knowledge | 5.41[g] (1.45) |
| 9. Understanding | 5.36[e] (1.32) |
| 10. Capacity to comply | 5.46[h] (1.20) |
| 11. Impulsivity | 5.16[c] (1.34) |
| 12. Negative emotions | 5.01[i] (1.36) |
| 13. Opportunity to violate | 5.00[j] (1.29) |

Means with differing superscripts indicate significant differences at $p < .004$ (Bonferroni correction).

Table 5. Empirical understanding of compliance: Observed associations (bivariate correlations) of compliance mechanisms with compliance according to empirical analysis.

| | Compliance | Costs of compliance | Benefits of compliance (perceived threat) | Punishment likelihood | Punishment severity | Social norms | Moral alignment | Perceived effectiveness and proportionality | Duty to obey the law | Knowledge | Understanding | Capacity to comply | Impulsivity | Negative emotions |
|---|---|---|---|---|---|---|---|---|---|---|---|---|---|---|
| **Costs of compliance** | −.06 | | | | | | | | | | | | | |
| **Benefits of compliance (perceived threat)** | .51 | .03 | | | | | | | | | | | | |
| **Punishment likelihood** | .06 | .14 | .08 | | | | | | | | | | | |
| **Punishment severity** | −.04 | .17 | −.01 | .08 | | | | | | | | | | |
| **Social norms** | .43 | −.09 | .20 | .14 | −.03 | | | | | | | | | |
| **Moral alignment** | .61 | −.06 | .65 | .04 | −.07 | .29 | | | | | | | | |
| **Perceived effectiveness and proportionality** | .58 | −.12 | .64 | .03 | −.10 | .31 | .76 | | | | | | | |
| **Duty to obey the law** | .35 | −.09 | .31 | .00 | −.10 | .16 | .37 | .36 | | | | | | |
| **Knowledge** | .16 | .05 | .15 | .08 | .02 | .09 | .15 | .14 | .12 | | | | | |
| **Understanding** | .35 | −.06 | .25 | −.01 | −.00 | .23 | .35 | .37 | .28 | .21 | | | | |
| **Capacity to comply** | .66 | −.12 | .36 | .08 | −.05 | .55 | .47 | .47 | .30 | .13 | .38 | | | |
| **Impulsivity** | −.24 | .07 | −.13 | .06 | .04 | −.11 | −.15 | −.14 | −.22 | −.07 | −.13 | −.16 | | |
| **Negative emotions** | −.16 | .32 | −.09 | .07 | .17 | −.13 | −.18 | −.21 | −.22 | −.04 | −.15 | −.20 | .23 | |
| **Opportunity to violate** | −.27 | .08 | −.16 | −.02 | .04 | −.17 | −.16 | −.16 | −.11 | −.05 | −.02 | −.22 | .14 | .07 |

opportunities for offending (opportunity theories). As such, compliance was greater the more that participants saw benefits of compliance (i.e., perceived the virus as threatening), saw others around them comply, morally agreed with social distancing measures, saw these measures as effective and proportional, felt a general duty to obey the law, had knowledge of social distancing measures, understood these measures, and were practically able to comply with them. Conversely, compliance was lower the more that participants were impulsive, experienced negative emotions, or saw opportunities to violate social distancing measures. Practical capacity to comply (from capacity theories) showed the strongest association with compliance ($r = .66$), while punishment severity (deterrence from rational choice theories) showed the weakest ($r = -.04$). These associations concord with previously observed patterns of the relative associations of these mechanisms with compliance in research using the same materials (see S4 File).

### Comparing lay understanding and empirical evidence

Because the empirically observed associations only exist at the aggregate level (as a set of correlation coefficients derived from the sample as a whole), it is not possible to directly compare these to lay perceptions of these associations among individual participants. Specifically, while the latter vary between individuals (such that particular mechanisms may be perceived as more or less influential by different participants—as indicated by the standard deviations in Table 4), the former are constant for all participants (i.e., a single correlation coefficient derived from the sample as a whole), and hence lack the variation required for comparison at the individual level. To compare lay perceptions of how the different compliance mechanisms shape compliance with the empirically observed associations as shown by the correlation results, we therefore followed the approach of Willoughby et al. [46], by aggregating individual-level perceptions into sample-level mean scores for each of the 13 compliance mechanisms. The resulting mean scores indicate how influential each of these mechanisms was on average perceived to be for shaping compliance. Because the aggregated perceived associations exist at the sample level, these can be directly compared with the (sample-level) empirically observed associations. Following the approach of Willoughby et al. [46], the resulting mean scores were transformed to a scale between 0 and 1 (such that 0 was equivalent to "disagree completely" (i.e., a Likert score of 1), and 1 was equivalent to "agree completely" (i.e., a Likert score of 7)), to situate them on the same scale as the (absolute value of the) correlation coefficients. In this way, the overall concordance between lay perceptions and empirically observed associations across the different mechanisms can be assessed.

Fig 1 displays the empirically observed associations between compliance and the different compliance mechanisms (as indicated by the absolute value of their correlations), as well as the (mean) perceived strength of these associations as perceived by lay participants. The figure shows that participants on average recognized the strong influence of several behavioral mechanisms (e.g., benefits of compliance, moral alignment with the measures, practical capacity to comply); moreover, they also recognized the more limited influence of other mechanisms, such as deterrence.

Nevertheless, there also were considerable discrepancies between lay and empirical understanding of these associations. To begin with, Fig 1 shows that participants on average seemed to perceive all the different compliance mechanisms as relatively influential for compliance, in contrast with the considerable variation that existed in the empirically observed associations. Indeed, on average, participants rated all mechanisms above the scale midpoint (0.5) in terms of perceived influence, even those that in fact showed only very weak empirical associations with compliance (e.g., knowledge of the rules, costs of compliance, punishment). Moreover, (mean) lay perceptions of the influence of the different compliance mechanisms showed relatively little variability, compared to the substantial differences between the strength of the empirically observed associations. To evaluate this, we computed the Coefficient of Variation [CV, see 185], which represents a standardized measure of dispersion, for both the (mean) perceived associations and the empirically observed associations. Comparison of this index indicated that (mean) lay perceptions of the influence of the different compliance mechanisms showed substantially less variability ($CV_{intuitions} = 6.29$) than the empirically observed associations, as indicated by the (absolute) correlations ($CV_{correlations} = 60.84$). In sum, on average, participants perceived the influence of

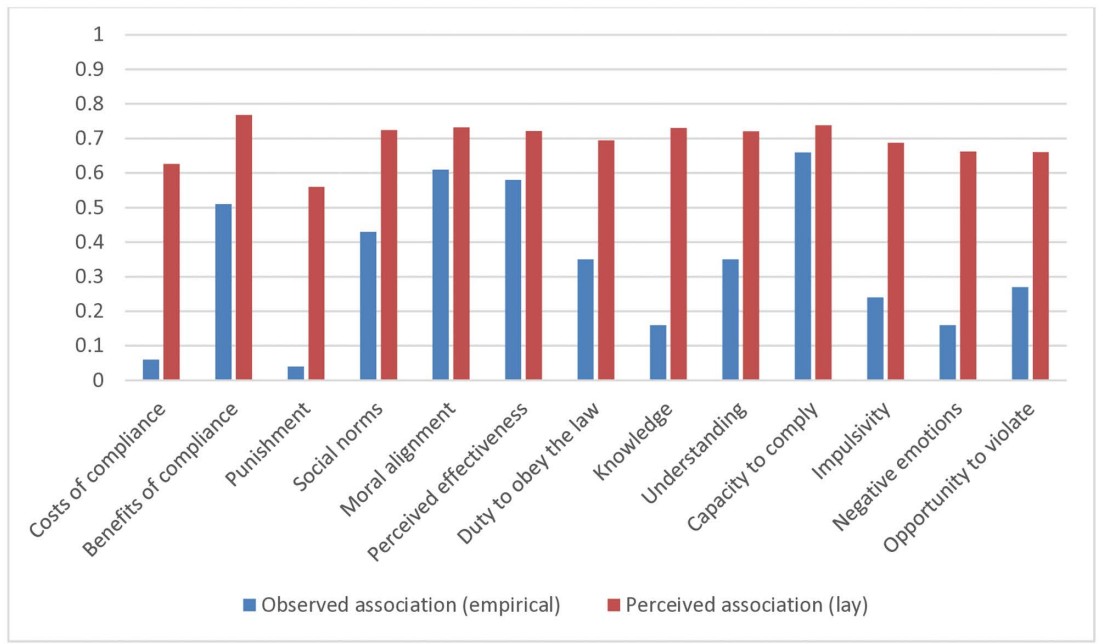

**Fig 1. Comparing lay and empirical understanding of compliance: Observed empirical associations (absolute correlations) of compliance mechanisms with compliance versus (aggregated) perceived associations with compliance according to lay participants.**

the different compliance mechanisms on compliance to be relatively strong and low in variability, which contrasted with the more modest and variable associations that were empirically observed.

Evidently, the differences between (mean) lay perceptions of the influence of these mechanisms and the observed empirical associations in part originate from differences in their measurement. For this reason, it is also informative to examine their overall concordance in terms of ranking. Instead of comparing the absolute strength of the association between the different mechanisms and compliance, this approach examines which mechanisms are most to least influential. Specifically, we examine whether the way that these mechanisms are on average ranked by lay participants in terms of their perceived influence on compliance may concord with their ranking according to the empirically observed associations. More concretely, this approach evaluates whether mechanisms that are (on average) seen as most to least influential according to lay participants may also show the strongest to weakest empirically observed associations with compliance.

Fig 2 illuminates this by plotting (mean) lay perceptions of the influence of the different mechanisms on compliance against actually observed correlations. Fig 2 demonstrates that there generally was considerable concordance between which mechanisms lay participants on average regarded as more (or less) influential and the empirically observed correlations between those mechanisms and compliance. On average, practical capacity to comply, moral alignment, and benefits of compliance were all predicted to have relatively strong influence on compliance, and indeed were among the mechanisms observed to have relatively stronger actual correlations with this outcome. Conversely, deterrence and costs of compliance were predicted to have a relatively weak influence on compliance, and indeed were among the mechanisms observed to have relatively weaker correlations. Generally, the figure displays a clear linear trend in which mechanisms that on average were perceived to be more influential also showed stronger empirical associations with compliance (and vice versa). This is confirmed by computing the Spearman rank correlation between (mean) lay perceptions (i.e., the average perceived influence of each of the 13 mechanisms) and the actually observed (absolute) correlations between

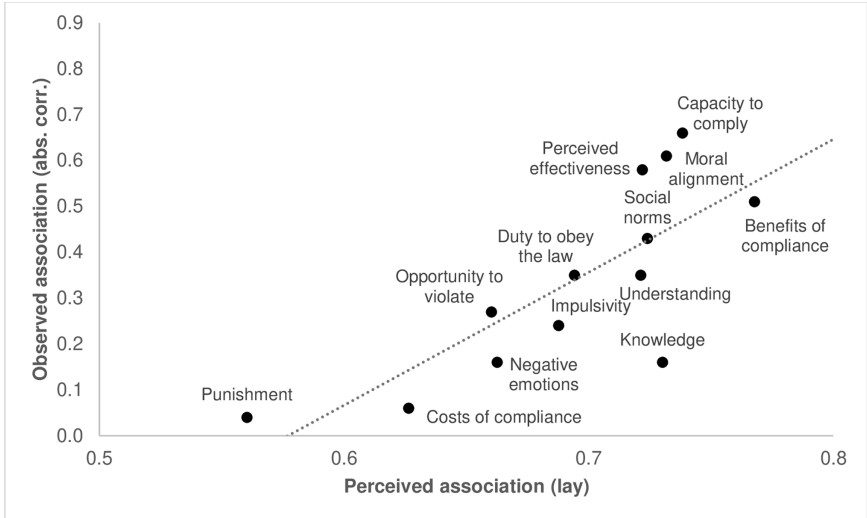

**Fig 2. Comparing lay and empirical understanding of compliance: Observed empirical associations (absolute correlations) of compliance mechanisms with compliance versus (aggregated) perceived associations with compliance according to lay participants (lay understanding).**

these mechanisms and compliance [cf. 46]: doing so revealed a very large correlation ($r$=.81). While this indicates strong overall concordance between lay understanding and empirical understanding, the figure does show some noteworthy deviations between the two. For example, while lay participants on average predicted the benefits of compliance (in terms of the perceived threat of the virus) to be the most influential for compliance, its actual correlation with compliance was ranked lower than those of other influential mechanisms, such as practical capacity to comply and moral alignment. On average, lay participants also perceived knowledge of the rules to have a relatively strong impact on compliance, while empirically, its correlation with compliance was relatively modest, and ranked among those mechanisms with the weakest associations.

## Policy preferences

Our final research question concerns how people's lay understanding of what shapes compliance may inform their preferences for policies to promote compliance. Table 6 indicates that participants most preferred policies that leveraged substantive legitimacy, by convincing people more that social distancing is effective and proportional. Conversely, they least preferred policies that leveraged deterrence (i.e., punishing people more to instill more fear of violating the measures). A fixed-effect (within) regression indicated that mean preference differed significantly between the different policies, $F(12, 39695) = 592.66$, $p < .001$.

Fig 3 displays participants' (mean) preferences (average preference for each of the 13 policies, again recoded to a scale between 0 and 1), and compares these to their (mean) lay perceptions of the influence of the corresponding mechanisms (average perceived influence of each of the 13 mechanisms, as detailed in the previous section). In general, participants' (mean) policy preferences appeared to track their (mean) perceptions of the influence of the underlying mechanisms: policies that leveraged mechanisms that were on average perceived as influential for shaping compliance (e.g., benefits of compliance) on average seemed to be preferred more; conversely, policies that leveraged mechanisms that were on average deemed less influential (e.g., deterrence) were on average preferred less.

Again, to evaluate their overall concordance, it is informative to compare their rankings. Fig 4 demonstrates this by plotting (mean) lay perceptions against (mean) policy preferences. The figure generally shows considerable concordance

**Table 6. Lay understanding of compliance: Preference for policies leveraging main compliance mechanisms to promote compliance.**

| Mechanism | Mean (SD) |
|---|---|
| 1. Costs of compliance | 5.22 (1.45) |
| 2. Benefits of compliance (perceived threat) | 5.63[a] (1.50) |
| 3. Punishment | 3.78[b] (1.94) |
| 4. Social norms | 5.33[c] (1.42) |
| 5. Moral alignment | 5.37[d] (1.50) |
| 6. Perceived effectiveness and proportionality | 5.73[e] (1.36) |
| 7. Duty to obey the law | 5.08[f] (1.65) |
| 8. Knowledge | 5.12[g] (1.63) |
| 9. Understanding | 5.21 (1.60) |
| 10. Capacity to comply | 5.53[h] (1.36) |
| 11. Impulsivity | 4.73[i] (1.66) |
| 12. Negative emotions | 5.27[j] (1.39) |
| 13. Opportunity to violate | 4.73[i] (1.71) |

Means with differing superscripts indicate significant differences at $p < .004$ (Bonferroni correction).

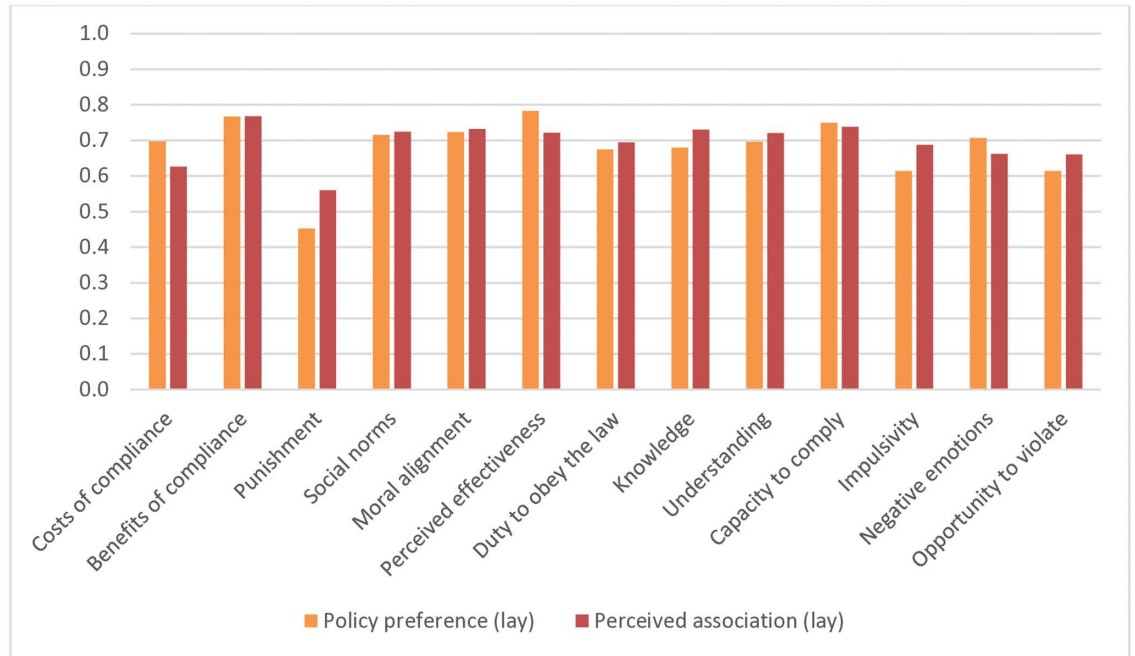

**Fig 3. Comparing lay understanding and policy preference: (Aggregated) perceived associations of mechanisms with compliance according to lay participants versus (aggregated) preference for policies to promote compliance.**

between (mean) lay perceptions of influence and (mean) policy preferences: if mechanisms were on average perceived to be more influential, policies that leveraged these on average also tended to be ranked among the most preferred. Conversely, if mechanisms were on average seen as less influential (especially deterrence), policies that leveraged these on

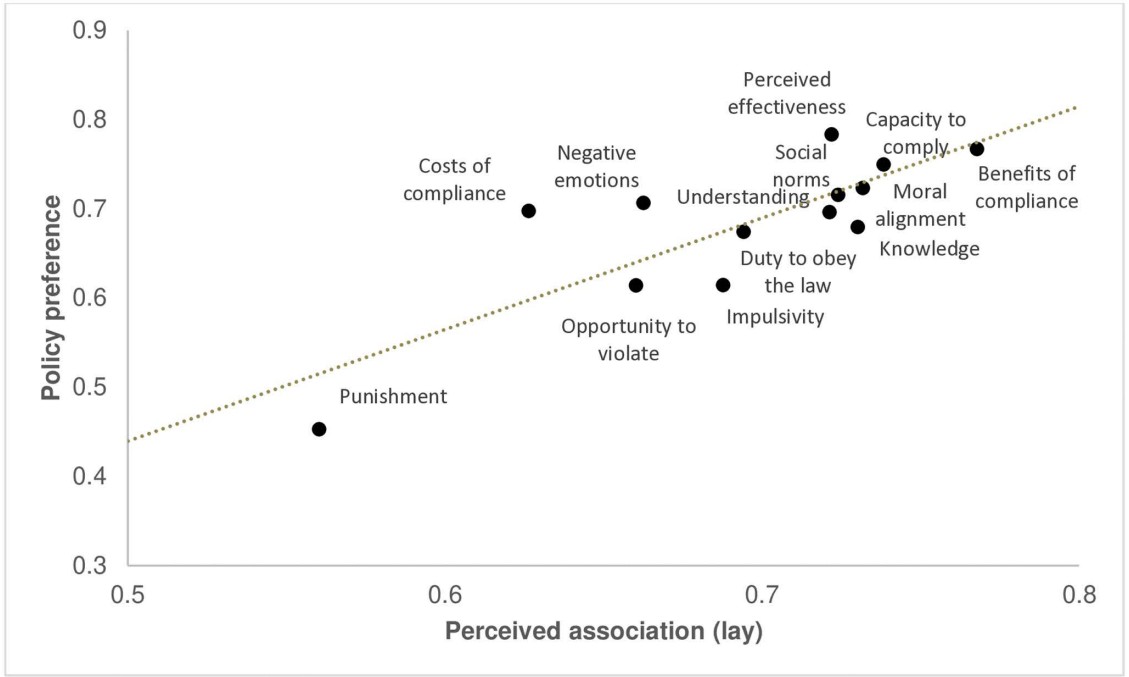

**Fig 4.** (Aggregated) preference for policies to promote compliance versus (aggregated) perceived effectiveness for promoting compliance (lay understanding).

average tended to be ranked among the least preferred. This was confirmed by computing the Spearman rank correlation between (mean) lay perceptions and (mean) policy preferences: this revealed a very large correlation ($r = .71$). Nevertheless, the figure also shows some noteworthy differences between the sample's lay perceptions and policy preferences. To begin with, while lay participants on average saw benefits of compliance as the most influential mechanism, it was (improving) substantive legitimacy (by convincing citizens that measures are effective and proportional) that was on average most preferred as a policy for shaping greater compliance. Similarly, while participants on average considered knowledge of the rules to be one of the more influential mechanisms for shaping compliance, it on average ranked among the less preferred policies for doing so. Conversely, among the mechanisms that lay participants on average saw as less influential, (reducing) negative emotions and costs were nevertheless on average ranked among the more preferred policies for enhancing compliance. In sum, the analysis demonstrates that at the aggregate level, there was substantial overall concordance between participants' lay perceptions of the mechanisms that shape compliance and the policies they preferred for promoting it: mechanisms that were on average seen as more influential on average were also preferred more as policies (and vice versa).

An important unresolved question, however, is whether the overall concordance between lay perceptions and policy preferences that exists at the aggregate level (such that mechanisms that are on average perceived as more effective are also on average preferred more in policies) may also exist at the individual level: do participants' lay perceptions of the influence of a particular mechanism on compliance also correspond with their preferences for leveraging that same mechanism in policy? To answer this question, we must look beyond the overall concordance between (aggregated) lay perceptions and (aggregated) policy preferences at the level of the sample, and zoom in on the concordance between lay perceptions and policy preferences for specific mechanisms at the level of the individual. For this purpose, we computed for each mechanism the correlation between participants' lay perception of its influence on compliance and their

preference for leveraging it in policy. Because both sets of variables were measured at the individual level (in contrast to the empirically observed associations), this analysis can directly illuminate whether lay perceptions of the influence of a particular mechanism may inform policy preferences for leveraging that same mechanism to promote compliance.

As the diagonal in Table 7 shows, correlations between lay perceptions of the influence of particular mechanisms and preferences for leveraging these in policy varied from small/moderate (.16, deterrence) to large (.35, social norms). As such, when zooming in on specific mechanisms, there was substantially less concordance between lay perceptions of their influence on compliance and policy preferences for leveraging them to promote compliance. Participants' lay perceptions of the influence of a particular mechanism on compliance did not automatically translate to a strong preference for policies that leveraged that mechanism; indeed, the perceived influence of some mechanisms was more strongly correlated with policies that leveraged an entirely different mechanism (e.g., increasing substantive legitimacy, benefits of compliance, self-control), while preferences for some policies were associated with the perceived influence of multiple mechanisms (e.g., increasing practical capacity to comply, or social norms for compliance). In sum, at the individual level, and when zooming in on specific mechanisms, the relationship between lay perceptions and policy preferences was much weaker and more diverse, compared to the high overall concordance that was observed across mechanisms at the aggregate level.

## Discussion

Citizens' lay understanding of behavior is a critical, but until now overlooked, element of how scientifically informed policies can come to shape behavior. The present research demonstrated that at the aggregate level, participants' lay understanding of what shapes compliance showed substantial concordance with empirical evidence on this question. More specifically, across the sample, their lay perceptions on which mechanisms were more or less influential were broadly aligned with empirically observed associations—although they seemed to overestimate these associations in absolute terms, and underestimate their variability. Moreover, at the aggregate level, their lay understanding of the influence of these mechanisms showed strong concordance with their preferences for policies that leveraged these mechanisms to increase compliance. These findings suggest that collectively, lay participants had considerable insight into what shapes compliance, which appeared to inform their policy preferences.

While these findings thus seem to show fertile soil for scientifically informed policy in this domain, one important caveat should be made. Although the present findings did indicate substantial concordance between lay and scientific perspectives, such concordance was principally observed at the aggregate level: when participants' lay understanding of the mechanisms that shape compliance were aggregated across the sample (and thus could be directly compared to the empirically observed associations between compliance and these mechanisms, which existed only at this aggregate level). However, associations observed at the aggregate level may not be directly indicative of associations at the individual level, where such associations may be weaker and show variation between individuals [65,186]. This point is also demonstrated by the association between lay understanding and policy preferences, which could be examined both at the individual and the aggregate level. At the individual level, participants' lay perceptions of the influence of specific mechanisms on compliance showed substantial variability (see Table 4), and for the most, showed only small to moderate concordance with their preferences for leveraging these mechanisms in policy to promote compliance (see Table 7). Accordingly, the observed high levels of concordance between lay and empirical understanding observed at the aggregate level need not indicate that individual participants were as capable of predicting the influence of the different mechanisms, or were able to do so in equal measure. Rather, it seems likely that underneath the high levels of alignment observed at the aggregate level, a more complex and diverse image may exist at the individual level, where some individuals may considerably under- or overestimate the influence of certain mechanisms on compliance, compared to an empirical understanding of this [for a similar conclusion, see 65]. The present data, however, preclude a more direct examination of such alignment at the individual level.

**Table 7. Comparing lay understanding and policy preference: Observed associations (bivariate correlations) between perceived (lay) association of mechanism with compliance and preference for policy to promote compliance.**

| Perceived association (lay) | Policy preference | | | | | | | | | | | | |
|---|---|---|---|---|---|---|---|---|---|---|---|---|---|
| | Costs of compliance | Benefits of compliance (perceived threat) | Punishment | Social norms | Moral alignment | Perceived effectiveness and proportionality | Duty to obey the law | Knowledge | Understanding | Capacity to comply | Impulsivity | Negative emotions | Opportunity to violate |
| Costs of compliance | **.17** | −.01 | .04 | .06 | .02 | .03 | .07 | −.00 | .01 | .09 | .03 | .15 | .10 |
| Benefits of compliance (perceived threat) | .16 | **.16** | .06 | .25 | .18 | .19 | .25 | .16 | .15 | .25 | .10 | .20 | .13 |
| Punishment | .12 | .10 | **.12** | .12 | .07 | .07 | .09 | .13 | .12 | .10 | .09 | .10 | .07 |
| Social norms | .18 | .16 | .06 | **.35** | .26 | .25 | .31 | .19 | .19 | .30 | .15 | .21 | .16 |
| Moral alignment | .20 | .15 | .07 | .27 | **.27** | .22 | .33 | .17 | .17 | .28 | .11 | .21 | .16 |
| Perceived effectiveness and proportionality | .21 | .24 | .11 | .34 | .28 | **.31** | .37 | .23 | .22 | .32 | .20 | .26 | .19 |
| Duty to obey the law | .21 | .12 | .07 | .27 | .24 | .21 | **.33** | .17 | .16 | .29 | .11 | .23 | .17 |
| Knowledge | .18 | .16 | .04 | .23 | .16 | .19 | .21 | **.28** | .24 | .18 | .13 | .15 | .08 |
| Understanding | .18 | .18 | .18 | .25 | .20 | .21 | .25 | .27 | **.26** | .21 | .16 | .19 | .08 |
| Capacity to comply | .22 | .11 | .06 | .26 | .20 | .17 | .27 | .16 | .16 | **.33** | .11 | .22 | .17 |
| Impulsivity | .18 | .13 | .07 | .27 | .21 | .20 | .26 | .16 | .13 | .29 | **.16** | .21 | .18 |
| Negative emotions | .16 | −.02 | .03 | .07 | .02 | .02 | .06 | .00 | .01 | .09 | .01 | **.22** | .04 |
| Opportunity to violate | .13 | .09 | .06 | .15 | .10 | .09 | .13 | .10 | .10 | .20 | .10 | .14 | **.18** |

Boldface indicates association between perceived association (lay) and policy preference for same mechanism.

## Theoretical implications

The present research provides a first empirical perspective on citizens' lay understanding of behavioral influence, focusing on the processes that shape compliant behavior. In this way, it provides an important complement to the existing literature on the translation of science to practice, such as that in translational or implementation science [187–191]. While existing work in this domain has extensively studied ways in which scientific knowledge may be implemented in practice to achieve behavioral change [192–196], it typically has not considered the way that citizens' own lay understanding of these processes may feature in this. The present research demonstrates that people's lay understanding of compliance relates not only to the processes they perceive to shape compliant behavior, but also informs (in some measure) the policies they prefer to achieve this. This demonstrates that lay understanding may be an important, but until now overlooked link between scientifically informed policies and their behavioral outcomes—which may potentially help to explain whether and why particular policies will be met with acceptance or reactance [195]. The present findings therefore underline that research into behavior change would benefit from greater attention to citizens' own lay understanding of these processes, and the way that this may shape their response to policies and interventions. Furthermore, this suggests that finding ways to navigate possible misalignment between lay and scientific understanding may be an important topic both for scientific research and scientifically informed policy.

By illuminating citizens' lay understanding of these processes, our findings also importantly extend the literature on compliance. This literature, which spans different academic domains and fields focusing on different behaviors and mechanisms, has sought to understand the processes that contribute to compliant and rule breaking behavior in both individuals and organizations [69]. Until now, however, little was known of how these and other relevant actors themselves understand these processes to operate, or the extent to which their lay understanding of this aligns with the scientific literature on compliance (with the exception of isolated studies focusing on different topics that have tangentially touched upon this question, e.g., [81,101–103]). The present research is the first to study this systematically. It demonstrates that within the studied setting, individuals collectively showed a nuanced understanding of the processes that shaped compliant behavior, which substantially aligned with empirical evidence on this question. While future research should determine if such alignment extends to other behaviors and settings, the current findings seem to suggest that lay individuals—on whose behaviors and responses the academic understanding of compliance is largely based—have, at least at the collective level, a complementary understanding of compliance. This suggests that lay understanding of compliance may provide a springboard, instead of an obstacle, for the dissemination and implementation of complementary academic knowledge. This also suggests that lay understanding may represent an important, untapped reservoir of experiential knowledge [197] that academic research may tap into to understand compliance in other settings.

Finally, the present findings also contribute to the (patchworked) global literature on lay understanding. This literature, which spans a range of academic disciplines, has generally indicated substantial differences between lay and scientific understanding in fields such as physics [54,56,80], mathematics [57], economics [58,60], and nutrition [28,198]. The high levels of alignment between lay and scientific understanding of what shapes compliant behavior observed here at the aggregate level present a striking contrast with this. By demonstrating such alignment, the present findings align with (sparse) research that has indicated global concordance between lay and scientific insight in particular domains, including parenting [40], behavioral consistency [93], memory [27], genetics [46], and biology [199].

An important question is what may explain why lay understanding aligns with scientific understanding in some domains, and differs from this in others. One possibility may be that laypersons possess better understanding of particular processes due to their direct personal experiences. For example, their direct experience with influencing others, or being influenced by them, may enable laypersons to develop a nuanced understanding of processes involved in compliance or parenting. Moreover, such processes chiefly relate to individual behaviors with observable outcomes and comparatively brief timespans; i.e., processes occurring at the meso level of complexity—the level at which according to Kelley [24] lay understanding is most likely to be extensive and valid (referring to commonsense psychology). This may explain why

more limited lay understanding has been observed of processes from other domains of scientific knowledge, including the laws of physics [54–56,80], mathematical principles [57], the workings of the economy [58,60], and nutritional processes operating in the body [28,198]—which typically are less about observable behavior, are less familiar or traceable, and are often situated at the micro or macro level of complexity. Yet while intuitively plausible, this explanation contrasts with findings that people often fail to recognize important influences on their behavior or that of others [e.g., social norms, 138], overestimate the role of particular factors, like personality [94,200], and base their views on what is effective on their moral or political preferences [201].

Another explanation for differing findings on the alignment of lay understanding with scientific understanding relates to the depth of understanding. It is possible that beyond the high levels of concordance that were observed at the global level in the present study, lay and scientific understanding become more misaligned at deeper levels of understanding. Most studies into lay understanding have focused on factual knowledge, for example about relevant processes in compliance (as in the present study), the economy [e.g., 83], or nutrition [e.g., 198]. However, few studies have zoomed in on *how* laypersons understand these processes to operate concretely. Most noteworthy is the body of work by diSessa [e.g., 54,80], which has examined people's lay understanding of processes in physics, by means of in-depth qualitative interviews. Such research demonstrates that although laypersons are familiar with important concepts like force and gravity, when asked to concretely explain how these processes operate (for example during a toss), their understanding comes to deviate substantially from that of science. These findings imply that although at the global level lay understanding may show substantial alignment with scientific understanding, this concordance may be reduced when zooming in on how particular processes concretely operate. Rather, their lay understanding at this level may reflect generalizations based on relatively simple abstractions [54], which may be permeated by personal preferences and beliefs [202], and lack the systematicity and integration of a scientific understanding. Regarding the present findings, this would imply that if laypersons would concretely explain how different behavioral mechanisms may come to shape compliance in a particular situation, the observed concordance with the empirical evidence might be reduced.

Finally, as noted earlier, differing findings on the alignment between lay and scientific understanding may be contingent on the level of analysis, such that alignment may be greater at the aggregate level than at the individual level. According to this explanation, lay understanding and scientific understanding may show more concordance at the aggregate level, where the (more diverse) perceptions of individual laypersons are integrated [a so-called wisdom-of-the-crowds-effect, see 65]. This possibility suggests that if lay understanding of compliance were assessed at the individual level (for example by means of an exam about empirically (dis)proven conclusions, see [81,102]), its alignment with the scientific understanding might be substantially reduced.

In sum, further, dedicated research is needed to evaluate how personal experience, level of complexity, depth of understanding, and level of aggregation may impact the level of alignment between lay and scientific knowledge. Future research may build upon the present research by relying on other methods to explore the level of alignment, and by zooming in more deeply on the way that laypersons concretely understand the relevant mechanisms to function. In this way, the present research provides an important first step towards a better and more systematic understanding of lay understanding, both of compliance and generally.

### Practical implications

In practical terms, the present findings demonstrate that when lay understanding is aligned with scientific knowledge, this may provide fertile ground for scientifically informed policies leveraging such knowledge. As the present findings demonstrated, policies that leveraged mechanisms that were perceived as influential were also preferred more, particularly when looked at from the aggregate level. This suggests that scientifically informed policies may gain more acceptance, and thereby more effectiveness, when these are aligned with citizens' shared understandings or beliefs about behavior. Accordingly, governments and companies who seek to translate behavioral scientific knowledge into practical

interventions could benefit from obtaining better insight into the lay understanding of targets of behavioral influence, and to tailor interventions to this where possible.

An important question for further study is how address instances where lay understanding is misaligned with scientific knowledge, and may present a barrier for scientifically informed policies—for example by giving rise to reactance [195]. Some perspectives have assumed that discrepancies between lay understanding and scientific understanding will naturally dissipate once such scientific knowledge is disseminated and becomes part of common knowledge [76,95]. However, for certain controversial topics, like climate change and vaccination, research shows that particular individuals reject the scientific consensus on these questions and maintain their personal convictions regarding their operation [35–38]. Such findings suggest that dissemination of the scientific understanding may not always suffice to dislodge lay understanding. Nonetheless, for many domains of knowledge, there are indications that educating people in the scientific understanding of these processes may increase the alignment between lay and scientific understanding.

Prior research on lay understanding, focusing specifically on public understanding of economics, suggests that education or training can promote economic literacy, which in turn can shape economic policy judgments [63,82]. Similar results have been observed for parenting knowledge [42] and for public understanding of genetics [203]. Such findings imply that in cases where lay understanding is misaligned with science, educating citizens about scientific knowledge could lead their lay understanding to become more aligned, and thereby possibly increase their receptiveness to scientifically informed policies. One indication of this that is directly relevant to the topic of compliance is provided by recent research by Kuiper et al. [204]. Specifically, these authors studied the effect of informing the public about empirical evidence about the deterrent effect of punishment [see 137,205]. Learning that science has found no conclusive empirical evidence that stronger punishment deters more (contrary to popular belief) significantly reduced participants' belief in the deterrent effect of punishment, as well as their preference for more punitive policies to reduce offending [for similar findings, see 206]. Such findings imply that communication of, or education in, relevant scientific evidence may be an important tool for reducing discrepancies between lay understanding and scientific knowledge. However, more systematic research is needed to evaluate this possibility, in light of evidence that people's personal (moral) views may shape their beliefs about what is effective [201], as well as their acceptance (or rejection) of empirical evidence that is aligned (or misaligned) with their views [108,207,208].

## Limitations and avenues for future research

Our findings are valuable by providing the first (to our knowledge) systematic analysis of people's lay understanding of what shapes compliant behavior, as well as its alignment with scientific evidence on this question. Nevertheless, we should also highlight some limitations to our approach, as well as ways in which future research might build on the present findings to deepen these.

A first limitation of our approach was that by asking participants to rate the influence of different compliance mechanisms identified in the academic literature, our measures may have shaped their thoughts about this. This also applies to the notion that earlier in the study, participants had answered questions about these mechanisms in relation to their own compliance. It is possible that our measures may thus have shaped participants' views on the relevance of these mechanisms, or attended them to mechanisms that they might not have identified spontaneously [209]. Also, this approach meant that participants were not able to spontaneously generate other mechanisms not identified in the academic literature on compliance. Our choice for this approach was motivated by our aim to compare participants' lay understanding of what shapes compliance with empirical evidence on this question, based on the core mechanisms identified in the academic literature on compliance [69,111,210]. Doing so necessitated that the same mechanisms were assessed both for the empirical associations and for lay understanding, so that the concordance between the two could be statistically estimated. Nevertheless, it would be worthwhile for future research to complement the present findings by utilizing qualitative approaches such as in-depth interviews, in which participants spontaneously generate and explain their beliefs about what shapes compliance [see 54].

A second limitation of our approach concerns the way in which lay understanding was measured. Our measure of this relied on Likert ratings of the perceived relevance of the different mechanisms for compliance, which we compared in terms of their pattern with that of the empirically observed associations (i.e., correlations). However, Likert scales are not directly comparable with correlations due to their distribution [211,212]; moreover, our scale labels were formulated in terms of agreement, which is not optimally suited for capturing the characteristics of correlations, particularly at the scale extremes (i.e., no association and perfect association). While our methods were suitable for comparing the global patterns of lay and empirical understanding, these limitations preclude a direct comparison between perceived and empirically observed associations in terms of their strength or magnitude. Future research may expand upon the present findings by using measures that focus more on the perceived strength with which mechanisms are associated with compliance than on their perceived relevance for this outcome. Moreover, such research may incorporate measures that better approximate the characteristics of correlations, such as slider measures [213–215]. Such measures of perceived association may also be suitable for conversion into Pearson's r statistics [216,217], thus enabling direct comparisons between lay and empirical understanding of compliance.

A third limitation concerns our focus on alignment at the aggregate (rather than the individual) level. As noted previously, while our current approach allowed us to assess the alignment between lay and scientific alignment of compliance at the aggregate level, this was not well suited to assessing such alignment at the individual level, or to evaluating the depth of people's understanding of these processes. We consider it likely, however, that people's lay understanding of compliance may show variation at the individual level, such that people may differ in which mechanisms they perceive to be more or less influential. Moreover, people likely will also vary in the depth of their understanding of these processes, such that they have different understandings of the way that particular mechanisms operate. For example, when reflecting about particular mechanisms, like punishment, individuals may not only recognize deterrent effects, but may also recognize more complex effects that have been identified in scientific research, like criminogenic effects, evasion, or adaptation [for a recent review, see 218]. Accordingly, it would be valuable for future research to zoom in on individual-level variability in lay understanding of compliance—both in terms of the mechanisms that people see as relevant, and the way in which they perceive these to operate. For this purpose, such research could rely on qualitative methods, which are both suitable for assessing depth and accuracy of lay understanding (e.g., [54,80]) and for identifying different belief profiles [e.g., see 219]. In this way, future research may build on the present findings by identifying different ideal types in individuals' lay understanding of compliance, as well as evaluating its alignment with the scientific understanding on a deeper, more substantive level.

A fourth limitation concerns the way that our research assessed participants' policy preferences. Because our study aimed to evaluate how participants' lay understanding of compliance would inform their preferences for policies to enhance it, we presented them with a series of policy interventions that leveraged the exact same compliance mechanisms that were used to assess their lay understanding of how compliance is shaped. Doing so enabled us to directly test how the perceived influence of a particular mechanism was associated with preferences for leveraging that mechanism to promote compliance. However, this approach means that participants were not able to propose other policies by which compliance in their view might be promoted. Moreover, our research operationalized these mechanisms into policies in a specific manner (e.g., in the case of knowledge of the rules: by making people more aware of the existence of this rule). However, it is possible that participants' policy preferences might differ if these were operationalized differently, for example in terms of a specific, concrete intervention (e.g., an information campaign). Furthermore, their policy preferences may reflect differences in practical feasibility, in that some mechanisms may be seen as influential (e.g., knowledge of the rules), but difficult to utilize (e.g., because levels of knowledge about social distancing rules already was substantial at the stage of the pandemic that the study was conducted), or vice versa. Accordingly, it would be valuable for future research to explore more deeply which policies participants spontaneously generate, and how these relate to their lay understanding of how compliance is shaped.

  

A fifth limitation concerns the way that the empirically observed associations were generated. These concerned associations between the different mechanisms and compliance based on self-reported measures. Self-reported data may be subject to response biases, however, such as social desirability bias [220]. Previous research does indicate that there can be strong concordance between self-reported and objective measures of compliance [221,222]. Moreover, the presently observed associations are globally in line with previous findings on the relevance of these mechanisms for compliance with social distancing measures, both from studies using the same materials (see S4 File) and more generally [167,223]. Future research may expand on the present findings by examining whether the concordance between lay and scientific understanding of compliance is replicated when other empirical indicators are used, including peer-reported or behavioral measures of compliance.

Finally, the present research does not illuminate to what extent the observed lay understanding of participants may extend beyond the present empirical setting, as well as its alignment with the empirical evidence and policy preferences. At present, our findings are restricted to the setting of mitigation measures against COVID-19 in the setting of the Netherlands, at the stage of the pandemic that our study took place. They do not yet illuminate participants' lay understanding of compliance in other settings, such as taxation, traffic, finance, or the environment, or for criminal offenses—or its alignment with science and policy there. Mitigation measures against COVID-19 were new and applied universally; compliance with these measures was often highly visible, and interventions to shape such behavior were widely implemented by governments. This contrasts sharply with compliance in other settings, where rules often have existed for a long time, and may not apply universally; where compliance may not be directly observable; and where interventions to shape behavior may be less concentrated. Because of these differences, it is possible that laypersons' understanding of what shapes compliance may be less well developed in these domains. However, it is also possible that people's lay understanding of compliance is rooted in more universal beliefs about behavioral influence, which transcend the specific setting of COVID-19 mitigation, and thus may also translate to different domains. As such, an important avenue for future research would be to explore people's understanding of compliance in other settings, such as taxation, traffic, finance, or the environment, or for criminal offenses. One particular challenge here is that behavioral science has not yet produced a full image of how the different mechanisms that have been identified in the compliance literature may shape compliant behavior in these settings. Rather, specific mechanisms have often been narrowly studied in context of a particular type of offending (e.g., legitimacy and opportunity in criminal offenses, see [224,225]; costs and benefits in financial crime, e.g., [133]; social norms in littering [226]) and tax evasion [227]). This underlines the need for more systematic research into the way that different compliance mechanisms may be at play in different settings [see 111]. Doing so is not only important as a benchmark for people's lay understanding, but also is essential for developing effective policies to enhance compliance in such settings, which may be obscured in narrow conceptualizations of compliance focusing on singular variables.

## Conclusion

Although behavioral insights increasingly inform policy, until now, little was known of how citizens themselves—the target of such interventions—think about shaping behavior, or how their lay understanding of this aligns with the scientific understanding. The present study demonstrates that at the aggregate level, citizens' lay understanding of how to shape compliant behavior showed considerable concordance with empirical evidence on this question (although they seem to overestimate the impact of mechanisms in absolute terms, and underestimate their variability). Moreover, their lay understanding informed their preference for policies that leverage the different mechanisms to increase compliance. Although this result does not indicate that laypersons individually had an accurate understanding of what shapes compliance, or typically agreed on the relevance of the different mechanisms, it does suggest that they possessed such an understanding collectively. In this way, their lay understanding may provide a stepping stone for scientifically informed policies that leverage empirical evidence on these processes. Moreover, these findings suggest that lay understanding can provide an important reservoir of experiential knowledge that the scientific study of these processes can tap into.

## Supporting information

**S1 File. Survey materials.**
(DOCX)

**S2 File. Dataset and syntax files.**
(DOCX)

**S3 File. Multiple imputation procedure, results, and checks.**
(DOCX)

**S4 File. Overview of previously observed associations with compliance per mechanism.**
(XLSX)

**S5 File. Standardized regression results.**
(DOCX)

## Acknowledgments

The authors thank Noah van Dongen for his thoughtful comments on earlier drafts of this paper.

## Author contributions

**Conceptualization:** Christopher P. Reinders Folmer, Malouke E. Kuiper, Benjamin van Rooij.

**Data curation:** Christopher P. Reinders Folmer.

**Formal analysis:** Christopher P. Reinders Folmer.

**Funding acquisition:** Benjamin van Rooij.

**Investigation:** Christopher P. Reinders Folmer, Benjamin van Rooij.

**Methodology:** Christopher P. Reinders Folmer, Benjamin van Rooij.

**Project administration:** Christopher P. Reinders Folmer.

**Resources:** Benjamin van Rooij.

**Software:** Christopher P. Reinders Folmer.

**Visualization:** Christopher P. Reinders Folmer.

**Writing – original draft:** Christopher P. Reinders Folmer, Benjamin van Rooij.

**Writing – review & editing:** Christopher P. Reinders Folmer, Malouke E. Kuiper, Benjamin van Rooij.

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
