## [Decision Letter · Decision Letter 0]

13 Aug 2025

Dear Dr. Reinders Folmer,

Thank you for submitting your manuscript to PLOS ONE. After careful consideration, we feel that it has merit but does not fully meet PLOS ONE’s publication criteria as it currently stands. Therefore, we invite you to submit a revised version of the manuscript that addresses the points raised during the review process.

This paper clearly has merit, and I am looking forward to the revised manuscript.I would like to see all reviewer comments addressed, but I would like to highlight the issue raised by Reviewer 2 around comparability of the statistics reported. Please address that point thoroughly.  

We look forward to receiving your revised manuscript.

Kind regards,

Johannes Schwabe

Academic Editor

PLOS ONE

Journal Requirements:

“This project was funded by a grant from ZonMw, the Netherlands Organisation for Health Research and Development (grant number 10430022010017), and by the European Research Council (ERC) under the European Union’s Horizon 2020 research and innovation programme (grant agreement no. 817680).”

Reviewers' comments:

Reviewer's Responses to Questions

**Comments to the Author**

1. Is the manuscript technically sound, and do the data support the conclusions?

Reviewer #1: Yes

Reviewer #2: Yes

2. Has the statistical analysis been performed appropriately and rigorously?

Reviewer #1: Yes

Reviewer #2: Yes

3. Have the authors made all data underlying the findings in their manuscript fully available?

Reviewer #1: Yes

Reviewer #2: Yes

4. Is the manuscript presented in an intelligible fashion and written in standard English?

Reviewer #1: Yes

Reviewer #2: Yes

Reviewer #1: This study addresses a significant gap in behavioral science and compliance literature by empirically examining lay understanding of behavioral mechanisms, a topic that has received limited attention despite its policy relevance. The use of a large, representative sample, multiple imputation for missing data, and strong statistical analyses (correlation, regression, rank-order comparisons) enhances the credibility of the findings.

The integration of five distinct theoretical frameworks provides a comprehensive lens for understanding compliance. The study offers actionable insights into how public beliefs may support or hinder scientifically informed policy, with implications for behavioral interventions and public communication strategies.

The manuscript is well-written, logically structured, and supported by clear tables and figures that effectively communicate the findings.

Suggestions for Improvement

- While the aggregate-level comparisons are compelling, the manuscript would benefit from deeper exploration of individual-level variability. Consider including clustering or latent class analysis to identify distinct lay belief profiles.

- The study is context-specific. The authors acknowledge this, but a brief discussion of how findings might translate to other domains (e.g., tax compliance, environmental behavior) would strengthen the manuscript.

- The study assesses perceived influence but not the depth or accuracy of lay conceptualizations. Qualitative follow-up studies could explore how laypeople understand the causal pathways of compliance mechanisms.

This is a well-executed and insightful manuscript that makes a valuable contribution to behavioral science and policy research. I recommend publication pending minor revisions that address the points above, particularly the discussion of individual-level variability and generalizability.

Reviewer #2: In this paper, the authors examine the correspondence between lay perceptions of the effects of various COVID restrictions on restriction compliance and the actual effects of these measures. The paper finds that members of the public think a variety of strategies for eliciting compliance successfully induce it, but do not distinguish much between tactics. As a result, while their perceptions of the effectiveness of different tactics corresponds in rank-order fashion to scientifically observed effects, perceptions of effectiveness often outweigh actual effectiveness. This is especially the case when one looks at the effect of policies over and above the effect of other policies. However, perceptions of effectiveness do not appear to be driven (much) by wishful thinking in that policy support corresponds in a fairly minor way to perceived policy effectiveness.

There is a lot to like about the paper. It is able to integrate a range of disparate literatures to inform specific predictions as to what informs lay theories of compliance. It also lays out a helpful theoretical framework that separates rational choice-based, social learning-based, legitimacy-based, capacity-based, and opportunity-based reasons for compliance. The results are easily connected with theory and directly answer the research questions at hand. My suggestions are thus relatively minor.

First, and most importantly, the authors compare absolute correlation coefficients (and later standardised regression coefficients) of actual effects of mechanisms on behaviour with lay perceptions of effects by rescaling the latter to span from 0-1. I am not convinced that this results in a 1-to-1 comparison effects. Namely, the endpoints of the scale for perceived effectiveness are 'disagree completely' and 'agree completely.' 'Agree completely' implies that a respondent thinks that there is an effect distinguishable from 0, but it need not imply a correlation of 1 - that an increase in use of the mechanism results in a perfect increase in compliance. 'Disagree completely' is a bit more murky, and could possibly mean 0 effect, or even a backfire effect. While it is too late to ask the respondents in this sample, one way to get a more accurate read is to see what kind of percentage change each tick on the scale corresponds to using an online sample of 100 or so. One can even convert these percentage changes into Pearson's r statistics, and thus for the correlations have a 1-to-1 comparison of perceived and actual correlations. This would also affect the analyses of perceived variability of effects as well.

Second, I am not sure that the comparison between perceived effects and standardised regression coefficients is fair. The perceived effectiveness question is inherently a bivariate prospect. In order to ask about perceived effectiveness in a way that corresponds to regression coefficients, one would need to ask something along the lines of "Over and above [insert mechanisms here], mechanism X would induce compliance." Since this was not asked, there isn't really a 1-to-1 comparison that can be done between perceived effects and regression coefficients. While for transparency's sake I can support the items remaining in an Appendix, they likely should not be in the main text.

Third, and tentatively, the authors mention in Lines 610-612 that they cannot estimate individual-level effects of policies and compare them to lay perceptions of effects. However, depending on whether the authors are willing to make some assumptions, Bayesian Causal Forests do estimate individual-level treatment effects, including with observational data (Caron et al. 2022). There is a package in R (bcf) that can run this kind of analysis. However, because the generated effects are based on assumptions that cannot be tested with real data, I can understand the authors being wary of this method.

Fourth, and finally, a small sentence as to which insight from the behavioural revolution the authors are referring to would improve the clarity of that first paragraph in the manuscript.

REFERENCES:

Caron, Alberto, Gianluca Biao, and Joanna Manolopoulou. 2022. "Shrinkage Bayesian Causal Forests for Heterogeneous Treatment Effects Estimation." Journal of Computational and Graphical Statistics 31(4): 1202-1214.

**Do you want your identity to be public for this peer review?** For information about this choice, including consent withdrawal, please see our Privacy Policy

Reviewer #1: **Yes: ** Gudberg K. Jonsson, Director of the Social Science Research Institute & Human Behavior Laboratory, University of Iceland

Reviewer #2: No

---

## [Author Response · Author response to Decision Letter 1]

21 Nov 2025

Christopher Reinders Folmer

Amsterdam Law School, University of Amsterdam

Department of Jurisprudence

Center of Law and Behavior

P.O. Box 15544

1001 NA Amsterdam

e-mail: c.p.reindersfolmer@uva.nl

Dr. Johannes Schwabe

Editor of PLOS One

Registry of Senior Australians (ROSA)

South Australian Health and Medical Research Institute

North Terrace, Adelaide, SA 5001

Australia

Topic: Revised version of our manuscript

Dear Dr. Schwabe,

With this letter, we send you the revised version of manuscript “The People versus Behavioral Science: Alignment between lay and scientific understanding of compliance” (PONE-D-25-28768).

We were grateful to receive the opportunity to revise our manuscript. We found the comments that were raised by you and by the Reviewers very helpful for improving it in terms of the contribution of our work, its limitations, and the avenues that it reveals for further research. In response, we have revised the manuscript to accommodate the suggestions. The manuscript now more clearly discusses the comparability of our measures of lay and empirical understanding, the question of individual-level variability and depth of understanding, and the relevance of our findings for compliance in other domains. Moreover, the revised manuscript better articulates how the present research is motivated by the ongoing “behavioral revolution” in practice, and omits the standardized regression results from the main text. As a result, we believe that the quality of the manuscript has much improved. We very much hope that you agree, and look forward to hearing from you.

We respond to all comments in detail below. We thank you for considering our manuscript for publication in your journal.

Yours sincerely,

The authors

PS: in response to your question about financial disclosure (additional requirements point 3), we hereby confirm that the funders had no role in study design, data collection and analysis, decision to publish, or preparation of the manuscript. 

Editor’s points

0. This paper clearly has merit, and I am looking forward to the revised manuscript.

We are happy to hear that the Editor viewed merit in our work, and thank them for their encouragement.

1. I would like to see all reviewer comments addressed, but I would like to highlight the issue

raised by Reviewer 2 around comparability of the statistics reported. Please address that point

thoroughly.

We thank the Editor for highlighting this point. Below, we address all points that were raised by the reviewers. We have made sure to address especially thoroughly Reviewer 2’s point on the comparability of the reported statistics (please see Reviewer 2 Point 1).

Additional requirements

Thank you for highlighting this. We have carefully checked the manuscript and files and believe that they now fully meet the journal’s requirements.

Thank you for attending us to this. We have removed all funding-related text from the revised manuscript.

As noted in the cover letter, we hereby confirm that the funders had no role in study design, data collection and analysis, decision to publish, or preparation of the manuscript.

4. When completing the data availability statement of the submission form, you indicated that you will make your data available on acceptance. We strongly recommend all authors decide on a data sharing plan before acceptance, as the process can be lengthy and hold up publication timelines. Please note that, though access restrictions are acceptable now, your entire data will need to be made freely accessible if your manuscript is accepted for publication. This policy applies to all data except where public deposition would breach compliance with the protocol approved by your research ethics board.

We confirm that all data and materials are now publicly accessible via our institution’s FigShare repository (please see S2 File Dataset and syntax files).

Thank you for highlighting this. In the revised manuscript, we have included a section on ethical approval and consent (p. 17), which provides these details. The manuscript does not include the original ethical approval forms (see attached files). Should the Editor desire for these documents to be included in a final submission, then we are happy to attach these to the manuscript as Supporting Information.

We confirm that all publications suggested by the reviewers have been reviewed and evaluated for relevance.

We have carefully reviewed the reference list to confirm that it is complete and correct. The manuscript (to our knowledge) does not cite any papers that have been retracted.

Reviewer 1’s points

0. This study addresses a significant gap in behavioral science and compliance literature by empirically examining lay understanding of behavioral mechanisms, a topic that has received limited attention despite its policy relevance. The use of a large, representative sample, multiple imputation for missing data, and strong statistical analyses (correlation, regression, rank-order comparisons) enhances the credibility of the findings. The integration of five distinct theoretical frameworks provides a comprehensive lens for understanding compliance. The study offers actionable insights into how public beliefs may support or hinder scientifically informed policy, with implications for behavioral interventions and public communication strategies. The manuscript is well-written, logically structured, and supported by clear tables and figures that effectively communicate the findings.

This is a well-executed and insightful manuscript that makes a valuable contribution to behavioral science and policy research. I recommend publication pending minor revisions that address the points above, particularly the discussion of individual-level variability and generalizability.

We thank the reviewer for their evaluation of our work, and hope that our response and revisions have sufficiently addressed their valuable suggestions.

1. While the aggregate-level comparisons are compelling, the manuscript would benefit from deeper exploration of individual-level variability. Consider including clustering or latent class analysis to identify distinct lay belief profiles.

We agree with the reviewer that further exploration of individual-level variability in lay understanding would be a valuable area of further exploration. We indeed consider it likely that lay understanding of compliance may vary between individuals, and that it is important to understand how. However, we assert that this question is better suited for follow-up research. The main purpose of the present research was to examine whether lay understanding about the mechanisms that shape compliance may align with empirical scientific evidence on these processes. As such, the study was designed to compare sample-wide statistical associations between these mechanisms and compliance (e.g. correlations) with aggregated perceptions of these associations across lay participants. The study design is concerned with the aggregate level (i.e., correlations computed across the whole sample), thus our data is not well suited to assessing variability in alignment at the individual level. Moreover, zooming in on individual-level variability would also raise questions about which individual-level factors may explain such variability—questions that the present research was not designed to provide answers to. We consider individual-level variability in lay understanding to be a highly important question for further exploration, which requires a research design that allows us to chart people’s individual differences. Instead of their understanding of the relative influence of different mechanisms, such a project might focus more on the depth of their understanding of the way that particular mechanisms operate, and how this aligns with the scientific understanding of these processes (e.g., see Kuiper, 2025). To do so, such research might utilize qualitative methods, which are suitable for assessing the depth and accuracy of lay understanding (see Reviewer 1’s point 3 below), and for distinguishing different belief profiles (e.g., see Kuiper, van Rooij, & Reinders Folmer, 2025). In the revised manuscript, we mention this as an important avenue for future research (p. 46-47):

A third limitation concerns our focus on alignment at the aggregate (rather than the individual) level. As noted previously, while our current approach allowed us to assess the alignment between lay and scientific alignment of compliance at the aggregate level, this was not well suited to assessing such alignment at the individual level, or to evaluating the depth of people’s understanding of these processes. We consider it likely, however, that people’s lay understanding of compliance may show variation at the individual level, such that people may differ in which mechanisms they perceive to be more or less influential. Moreover, people likely will also vary in the depth of their understanding of these processes, such that they have different understandings of the way that particular mechanisms operate. For example, when reflecting about particular mechanisms, like punishment, individuals may not only recognize deterrent effects, but may also recognize more complex effects that have been identified in scientific research, like criminogenic effects, evasion, or adaptation (for a recent review, see van Rooij, Kuiper, & Piquero, 2025). Accordingly, it would be valuable for future research to zoom in on individual-level variability in lay understanding of compliance—both in terms of the mechanisms that people see as relevant, and the way in which they perceive these to operate. For this purpose, such research could rely on qualitative methods, which are both suitable for assessing depth and accuracy of lay understanding (e.g., diSessa, 1993, 1998) and for identifying different belief profiles (e.g., see Kuiper, van Rooij, & Reinders Folmer, 2025). In this way, future research may build on the present findings by identifying different ideal types in individuals’ lay understanding of compliance, as well as evaluating its alignment with the scientific understanding on a deeper, more substantive level.

2. The study is context-specific. The authors acknowledge this, but a brief discussion of how findings might translate to other domains (e.g., tax compliance, environmental behavior) would strengthen the manuscript.

We agree with the reviewer that it would be valuable to know whether our sample’s lay understanding of what shapes compliance with COVID-19 mitigation measures might translate to other domains, including tax compliance and environmental behavior. This is, however, an empirical question, which the present study is unable to answer—not only because our study did not assess participants’ lay understanding of these other domains, but also because behavioral science has not yet produced a full image of how the different mechanisms that have been identified in the compliance literature may shape compliant behavior in these settings. Due to these limitations, it is at present impossible to provide a concrete answer as to whether the observed alignment between lay and scientific understanding of compliance in context of COVID-19 mitigation may translate to other domains.

Nonetheless, there are some concrete differences between these domains that may allow us to speculate about this question. To begin with, mitigation measures against COVID-19 were new and applied universally; compliance with these measures was often highly visible; and interventions to shape such behavior were widely implemented by governments. This contrasts sharply with domains like taxation and the environment, where many rules have existed for a long time, and may apply differently to different subsets of the population; where compliance is often not directly observable; and where there may be less of a concentrated effort to shape behavior. Based on these differences, it is possible that laypersons’ understanding of what shapes compliance in these domains will be less well developed than observed in this domain. However, it is also possible that people’s lay understanding in any of these domains is rooted in more universal beliefs about behavioral influence that transcend the specific setting of COVID-19 mitigation, and thus may apply also in different domains. This, however, is an empirical question, and one that only further research can answer. In the revised manuscript, we discuss this on pages 48-49:

At present, our findings are restricted to the setting of mitigation measures against COVID-19 in the setting of the Netherlands, at the stage of the pandemic that our study took place. They do not yet illuminate participants’ lay understanding of compliance in other settings, such as taxation, traffic, finance, or the environment, or for criminal offenses—or its alignment with science and policy there. Mitigation measures against COVID-19 were new and applied universally; compliance with these measures was often highly visible, and interventions to shape such behavior were widely implemented by governments. This contrasts sharply with compliance in other settings, where rules often have existed for a long time, and may not apply universally; where compliance may not be directly observable; and where interventions to shape behavior may be less concentrated. Because of these differences, it is possible that laypersons’ understanding of what shapes compliance may be less well developed in these domains. However, it is also possible that people’s lay understanding of compliance is rooted in more universal beliefs about behavioral influence, which transcend the specific setting of COVID-19 mitigation, and thus may also translate to different domains. As such, an important avenue for future research would be to explore people’s understanding of compliance in other settings, such as taxation, traffic, finance, or the environment, or for criminal offenses. One particular challenge here is that behavioral science has not yet produced a full image of how the different mechanisms that have been identified in the compliance literature may shape compliant behavior in these settings. Rather, specific mechanisms have often been narrowly studied in context of a particular type of offending (e.g., legitimacy and opportunity in criminal offenses, see Gill et al

---

## [Editor Report · Decision Letter 1]

25 Nov 2025

The People versus Behavioral Science: Alignment between lay and scientific understanding of compliance

PONE-D-25-28768R1

Dear Dr. Reinders Folmer,

We’re pleased to inform you that your manuscript has been judged scientifically suitable for publication and will be formally accepted for publication once it meets all outstanding technical requirements.

Kind regards,

Johannes Schwabe

Academic Editor

PLOS ONE

Additional Editor Comments (optional):

You have addressed all reviewer comments comprehensively, and the manuscript is in great shape. I am pleased to accept it for publication. Congratulations on an excellent paper.
---

## [Editor Report · Acceptance letter]

PONE-D-25-28768R1

PLOS One

Dear Dr. Reinders Folmer,

I'm pleased to inform you that your manuscript has been deemed suitable for publication in PLOS One. Congratulations! Your manuscript is now being handed over to our production team.

Kind regards,

on behalf of

Dr. Johannes Schwabe

Academic Editor

PLOS One